# Iron minerals inhibit the growth of *Pseudomonas brassicacearum* J12 via a free-radical mechanism: Implications for soil carbon storage

**Hai-Yan Du[1], Guang-Hui Yu[1, 2], Fu-Sheng Sun[1, 2], Muhammad Usman[3, 4], Bernard A. Goodman[5], Wei Ran[1], Qi-Rong Shen[1]**

1.  Jiangsu Provincial Key Lab for Organic Solid Waste Utilization, College of Resources & Environmental Sciences, Nanjing Agricultural University, Nanjing 210095, China

2.  Institute of Surface-Earth System Science, Tianjin University, Tianjin 300072, China

3. Environmental Mineralogy, Center for Applied Geosciences, University of Tübingen, 72074 Tübingen, Germany

4. Institute of Soil and Environmental Sciences, University of Agriculture, Faisalabad 38040, Pakistan

5. College of Physical Science and Technology, Guangxi University, Nanning 530004, Guangxi, China

*Correspondence to*: Guang-Hui Yu (yuguanghui@njau.edu.cn or yuguanghui@tju.edu.cn)

**Abstract.** Natural minerals in soil can inhibit the growth of bacteria that protect

organic carbon from decay. However, the mechanism inhibiting the bacterial growth

remains poorly understood. Here, using a series of cultivation experiments and

biological, chemical and synchrotron-based spectral analyses, we showed that

kaolinite, hematite, goethite and ferrihydrite had a significant inhibitory effect on the

25 growth of the model bacteria--*Pseudomonas brassicacearum* J12, which was more

prominent with a concentration of 25 mg mL$^{-1}$ than it was with either 10 mg mL$^{-1}$ or

5 mg mL$^{-1}$. In contrast, montmorillonite promoted the growth of J12. Compared to

Al-containing minerals, Fe(III)-containing minerals produced more hydroxyl radical

(HO$^{\bullet}$) that have high efficiency for the inhibition of J12. Moreover, a significant

positive correlation between HO$^{\bullet}$ radical and Fe(II) was found, suggesting that Fe(II)

contributes to the generation of HO$^{\bullet}$. Furthermore, both micro X-ray fluorescence

and X-ray photoelectron spectroscopies indicated that surface Fe(III) was reduced to

Fe(II) which can produce HO$^{\bullet}$ through the well-known Fenton reaction series.

Together, these findings indicate that the reduced surface Fe(II) derived from

35 Fe(III)-containing minerals inhibit the growth of *Pseudomonas brassicacearum* J12

via a free-radical mechanism, which may serve as an ubiquitous mechanism between

iron minerals and all of the heterotrophic bacteria in view of taxonomically and

ecologically diverse heterotrophic bacteria from terrestrial environments as a vast

source of superoxide.

**Keywords**: Fenton reaction; hydroxyl radical (HO$^{\bullet}$); inhibition of bacterial growth;

iron minerals; soil carbon storage

## 1. Introduction

A variety of minerals exhibit bacterial inhibition properties by releasing Al(III) or Fe(II) (Morrison et al., 2014; McMahon et al., 2016; Williams, 2017). Hence, natural minerals have long been used as bactericidal agents for human pathogens (Williams and Haydel, 2010; Williams et al., 2011). The bacterial inhibition property of a mineral is associated with the particular chemistry and with the mineral properties, resulting in the various bacterial inhibition mechanisms of minerals such as increase of membrane permeability and oxidative damage (Williams et al., 2008). Iron oxides are abundant in terrestrial and aquatic environments and exist predominantly as ferric minerals such as goethite, ferrihydrite, and hematite (Cornell and Schwertmann, 2003; Meunier, 2005; Chesworth, 2008). Due to the ubiquity of soil iron minerals and their distinct inhibition properties, which may affect soil carbon storage and nutrient turnover, investigations of the inhibitory potential of iron minerals on microorganisms are of great importance.

To better understand the inhibition of bacteria by minerals, the mineral type and size should be examined. Previous studies have demonstrated that Al(III)- and Fe(II)-containing minerals can inhibit the growth of bacteria (McMahon et al., 2016). For Al(III)-containing minerals, their toxicity mainly depends on the release of Al(III), an extensively toxic element to bacteria (McMahon et al., 2016). However, Fe(II)-containing minerals usually cause oxidative damage to bacteria, i.e., the oxidative role of reactive oxygen species (ROS), particularly by involving hydroxyl radicals (HO$^\bullet$) that are generated by an Fe(II) catalyzed Fenton reaction where Fe(II) reacts with hydrogen peroxide (H$_2$O$_2$) to form HO$^\bullet$ radicals (Stohs and Bagchi, 1995; Williams et al., 2011; Wang et al., 2017a; Usman et al., 2018). However, it is unclear

whether the common Fe(III)-containing minerals in soil have a similar inhibition activity with Al(III)- and Fe(II)-containing minerals.

Taxonomically and ecologically diverse heterotrophic bacteria from terrestrial environments are a vast source of superoxide ($O_2^{\bullet-}$) and $H_2O_2$ (Diaz et al., 2013). Meanwhile, Fe(III)-containing minerals can catalyze the decomposition of $H_2O_2$ to generate strong oxidizing ROS (predominantly $HO^{\bullet}$ radical) through Fenton-like reactions (equations 1-2) (Petigara et al., 2002; Garrido-Ramírez et al., 2010; Georgiou et al., 2015; Usman et al., 2016).

$$\equiv \text{Fe(III)-OH} + H_2O_2 \rightarrow \text{ } \equiv \text{Fe(II)} + H_2O + HO_2^{\bullet} \tag{1}$$

$$\equiv \text{Fe(II)} + H_2O_2 \rightarrow \text{ } \equiv \text{Fe(III)-OH} + HO^{\bullet} \tag{2}$$

where $\equiv \text{Fe(III)} - \text{OH}$ represents the iron mineral surface.

These Fenton-like reactions are well known as a type of heterogeneous catalysis (involving Fe minerals), which is distinct from homogeneous Fenton reactions (based on soluble Fe(II) in acidic media) (Garrido-Ramírez et al., 2010). The major advantage of heterogeneous catalysis is that it operates well over a wide range of pH values, while homogeneous catalysis displays optimal performance only at a pH of ~3 (Garrido-Ramírez et al., 2010). Furthermore, some researchers had demonstrated that surface Fe(II) was generated in the systems of $H_2O_2$ and ferric minerals (Kwan and Voelker, 2003; Polerecky et al., 2012). To date, the impact of Fe(III)-containing minerals on heterotrophic bacteria remains largely unexplored.

Here, we hypothesize that Fe(III)-containing minerals can inhibit the growth of heterotrophic bacteria through a free-radical mechanism (i.e., Fenton-like reactions).

To test our hypothesis, we designed a series of cultivation experiments to monitor the growth of the model bacteria--*Pseudomonas brassicacearum* J12--in the presence of various minerals and in a mineral-free control. Various minerals, including montmorillonite, kaolinite, hematite, goethite and ferrihydrite, were used as the model Al(III)- or Fe(III)-containing minerals because they are broad-based in a wide range of soils (Cornell and Schwertmann, 2003; Meunier, 2005; Chesworth, 2008). Specifically, montmorillonite and kaolinite are Al(III)-containing minerals, while hematite, goethite and ferrihydrite belong to Fe(III)-containing minerals. Meanwhile, *Pseudomonas brassicacearum* J12 was selected as the model heterotrophic bacterium because it represents a major group of rhizobacteria that aggressively colonize plant roots in soils (Zhou et al., 2012). In this study, the objectives were to 1) examine and compare the inhibition properties of Al and Fe minerals on J12; 2) build the correlation between solution chemistry and HO$^\bullet$ and the growth of J12; and 3) identify the mechanism by which Fe(III)-containing minerals inhibit J12. Throughout our experiments, the HO$^\bullet$ was trapped by terephthalic acid (TPA) (non-fluorescent), and the reaction's fluorescent product, i.e., 2-hydroxylterephthalic acid (HTPA) (Li et al., 2004), was quantitated in a high-performance liquid chromatography (HPLC) system. Correlative micro X-ray fluorescence (μ-XRF) and synchrotron-based Fourier transform infrared (SR-FTIR) spectroscopies were used to probe the *in situ* distribution and species of the Fe and extracellular polymeric substances (EPS), respectively (Luo et al., 2014; Sun et al., 2017a). X-ray photoelectron spectroscopy (XPS) was also used for analyzing the oxidation state(s) and speciation of Fe (Wilke et al., 2001; Yamashita and Hayes, 2008).

**2. Materials and Methods**

## 2.1. Mineral preparation

Five minerals were selected in this study, including kaolinite ($Al_2O_3.2SiO_2·2H_2O$, 98%, Aladdin Reagent Company, Shanghai, China), montmorillonite (($Al_2,Mg_3)Si_4O_{10}(OH)_2·nH_2O$, 98%, Aladdin Reagent Company, Shanghai, China) and synthetic hematite, goethite and ferrihydrite. All of the three iron minerals were synthesized by a previously described method (Schwertmann and Cornell, 2007). In brief, ferrihydrite was prepared by dissolving 40 g $Fe(NO_3)_3•9H_2O$ in 500 mL deionized water, and then 330 mL of 1 M KOH was added. Goethite was prepared by mixing 180 mL of 5 M KOH with 100 mL of 1 M $Fe(NO_3)_3•9H_2O$, and then the resulting mixture was aged for 60 h at 70 °C. Hematite was synthesized by mixing 2 L of 0.002 M $HNO_3$ (98 °C) with 16.16 g of $Fe(NO_3)_3•9H_2O$ and then aging for 7 d at 98 °C. Once prepared, all three suspensions were dialyzed with deionized water for 3 d to remove impurity ions, and then the pellets were air-dried. Powder X-ray diffraction (XRD) and FTIR analysis results for the used minerals are shown in Figs. S1-S2. All minerals were crushed and sieved through a 0.149 mm screen.

## 2.2. *Pseudomanas* cultivation experiments

The stock strain of J12 was inoculated in Nutrient Broth (NB) medium to an optical density ($OD_{600}$) of ~0.6. The NB medium includes beef extract, 3 g $L^{-1}$; Tryptone, 5 g $L^{-1}$; yeast extract, 0.5 g $L^{-1}$; Glucose, 10 g $L^{-1}$. The cultivation system contained 9.5 mL of NB medium and 0.5 mL of J12, with a concentration of minerals of 5, 10 or 25 mg $mL^{-1}$. The final pH of the cultivation system was adjusted to 7.2. Next, the

cultivation media were incubated for 12 h on a shaking incubator (180 rpm) at 28 °C. Then, 50 µL of the cultures were transferred to fresh medium (10 mL) so that the effects of minerals were negligible. Measurement of the $OD_{600}$ on mineral suspension was shown in Table S1. After 8 h growth, the growth of J12 was monitored by measuring $OD_{600}$ of the new culture and the photographs are shown as Fig. S3. The control experiment was performed without any mineral. All experiments were performed in triplicate. The particle size distribution of the applied raw minerals and the minerals after 12 h of incubation is listed in Fig. S4. According to the data provided by manufacturers, the specific surface area of kaolinite and montmorillonite are ~40 and 800 $m^2$ $g^{-1}$, respectively. The synthesis of hematite, goethite and ferrihydrite were referred to the method from Schwertmann and Cornell (2007), and their specific surface area are approximately 30, 20, 200-300 $m^2$ $g^{-1}$, respectively.

**2.3. HPLC analysis**

The $HO^{\bullet}$ was quantified in an Agilent 1260 Infinity HPLC system (Agilent Technologies, Inc., Germany) equipped with a Fluorescence Detector (G1321B) and a reverse-phase C18 column (Develosil ODS-UG5, 4.6 mm × 250 mm, Nomura Chemical Co., Japan). The mobile phase consisted of 200 mM $K_2HPO_4$ containing 2% of KCl (pH 4.37) and acetonitrile (90 : 10). Standard additions of 0, 0.05, 0.1, 0.5, and 1.0 µM HTPA were used to calibrate the HTPA response in each sample, with a linear response observed for all samples (Fig. S5). All experiments were

performed in triplicate.

**2.4. Correlative μ-XRF and SR-FTIR analysis**

After 12 h growth, the original culture of the 25 mg mL$^{-1}$ ferrihidrite treatment was frozen at −20 °C and directly sectioned without embedding. Then, thin sections (4 μm in thickness) were cut on a cryomicrotome (Cyrotome E, Thermo Shandon Limited, UK) and transferred to infrared-reflecting MirrIR Low-E microscope slides

(Kevley Technologies, Ohio, USA).

The SR-FTIR analysis was obtained at beam-line BL01B1 of the National Center for Protein Science Shanghai (NCPSS). Spectra were recorded in reflectance mode using a Thermo Nicolet 6700 FTIR spectrometer and a continuum infrared microscope with the following settings: aperture size 15 μm, step size 10 × 10 μm$^2$,

resolution 4 cm$^{-1}$, and 64 scans. Spectral maps were processed using Omnic 9.0 (Thermo Fisher Scientific Inc.). After baseline correction, map profiles of Fe-OH, C-H, C=O, C-N, and C-OH were created for peak heights at 3344, 2921, 1632, 1513, and 1030 cm$^{-1}$, respectively (Sun et al., 2017a and 2017b).

After SR-FTIR analysis, Fe image was collected at beamline 15U1 of Shanghai

Synchrotron Radiation Facility (SSRF) for the same region of the thin section. Fluorescence maps (μ-XRF) of Fe were obtained by scanning the samples under a monochromatic beam at E = 10 keV with a step size of 2.3 × 3.3 μm$^2$ and a dwell time of 1 s. Then, two positions were selected for Fe K-edge μ-X-ray absorption near-edge

structure (μ-XANES) analysis, and μ-XANES spectra were recorded using a 0.1 eV

step size with a Si drift detector. Standard samples of hematite, goethite, ferrihydrite,

iron(II) oxalate, and iron(III) oxalate were recorded in transmission mode. Iron(II)

oxalate and iron(III) oxalate represent organic complexing ferrous and ferric,

respectively, whereas hematite, goethite and ferrihydrite were used as the main iron

mineral species. Linear combination fitting of standards was also performed for the

180 μ-XANES spectra of samples, using ATHENA software (version 2.1.1). A standard

was considered to have a substantial contribution if it accounted for more than 10% of

a linear combination fit.

### 2.5. XPS analysis

The species of iron oxides were analyzed by XPS (PHI5000 Versa Probe,

ULVAC-PHI, Japan). All the samples were freeze-dried and ground to fine powders

prior to the XPS measurement. The XPS spectra were obtained with a

monochromatized Al $K\alpha$ X-ray source (1486.6 eV) and the pressure in the analytical

chamber was below $6 \times 10^{-8}$ Pa (Yangzhou University). For wide scan spectra, an

energy range of 0–1100 eV was used with the pass energy of 80 eV and the step size

of 1 eV. The high-resolution scans were conducted according to the peak being

examined with the pass energy of 40 eV and the step size of 0.06 eV. The precision

of XPS was 0.06 eV. In order to obtain the oxidation status of surface sites, narrow

scan spectra for Fe $2p_{3/2}$ were acquired. The carbon 1s electron binding energy

corresponding to graphitic carbon at 284.8 eV was used as a reference for calibration

purposes. Narrow scan spectra for Fe $2p_{3/2}$ were collected in binding energy forms and fitted using a least-squares curve-fitting program (XPSPEAK41 software). The XPS spectra were analyzed after subtracting the Shirley background that was applied for transition metals. The full width at half-maximum of those spectra was fixed constant between 1 and 3 and the percentage of Lorentzian–Gaussian was set at 20% for all the spectra.

## 2.6. Electron paramagnetic resonance (EPR) spectroscopy

The EPR spectra were recorded with a Bruker A300 X-band spectrometer (Guangxi University), which used a Gunn diode as microwave source and incorporated a high-sensitivity cavity. Individual spectra were recorded over scan ranges of 500 and 30 mT to observe the signals originating from transition metal ions and free radicals, respectively. Details of additional spectra and all other acquisition parameters are given in the references (Goodman et al., 2016). The $g$ values were calculated by reference to the Bruker ER4119HS-2100 marker accessory which has a $g$ value of 1.9800. Spectral data were processed using the Bruker WinEPR software; with samples recorded with the same values for the microwave power, modulation amplitude, time constant and conversion time, intensities were determined both from double integration of complete spectra after background correction, and the heights of individual peaks, and corrected for any differences in the receiver gain or number of scans. Simulations of spectra to test the validity of various models for the C-centre spectrum were performed using the Bruker SimFonia software.

## 2.7. Chemical analysis

At cultivation time of 2 h and 12 h of the original cultures, portions of the samples were centrifuged at 16,000 $g$ for 5 min, then filtered through a 0.45 µm membrane filter and analyzed with Inductively Coupled Plasma-Atomic Emission Spectroscopy (710/715 ICP-AES, Agilent, Australia) to detect the concentration of soluble Fe and Al. Total Fe and Fe(II) were determined with a modified 1,10-phenanthroline method (Amonette, 1998). Turbidity at 600 nm (a standard proxy for bacterial cell density) was measured using a Microplate Reader (Hach DR/2010) in mid-exponential phase. The pH of *Pseudomonas brassicacearum* J12 cultivated with different minerals or without mineral (control) was detected after 12 h. Eh of the suspension of minerals alone (25 mg ml$^{-1}$) and of bacteria-mineral mixture was detected by redox potentiometer (Orion star A211, Thermo Fisher scientific, USA). All experiments were performed in triplicate.

## 2.8. Statistical analysis

Significance was determined using one-way ANOVA followed by Tukey's HSD post hoc test, where the conditions of normality and homogeneity of variance were met; means ± SE (n = 3) that are followed by different letters indicate significant differences between treatments at $p < 0.05$. The one sample Kolmogorov-Smirnov test is used to test whether a sample comes from a specific distribution. In this study we used this procedure to determine whether the data set was normally distributed. In the regression equation, the parameters $R$ and $t$ represent

coefficient of determination and the result of *t*-test. Microsoft Excel (2010), Origin Pro8 and SPSS (18.0) were used for drawing the graphs and data analysis.

## 3. Results

### 3.1. Effect of mineral nature and their concentrations on J12 development

Compared to Control ($0.34 \pm 0.01$), the presence of montmorillonite significantly ($p < 0.05$) increased $OD_{600}$ (Fig. 1). Specifically, the $OD_{600}$ values of samples were $0.43 \pm 0.01$, $0.44 \pm 0.02$ and $0.43 \pm 0.01$ at the concentration of 5, 10 and 25 mg mL$^{-1}$, respectively. Presence of all other investigated minerals decreased $OD_{600}$ in the following order: ferrihydrite ($0.24 \pm 0.04$ and $0.09 \pm 0.01$) > goethite ($0.26 \pm 0.02$ and $0.14 \pm 0.00$) > hematite ($0.30 \pm 0.03$ and $0.16 \pm 0.02$) > kaolinite ($0.32 \pm 0.01$ and $0.20 \pm 0.01$) at 5 and 25 mg mL$^{-1}$, respectively, and ferrihydrite ($0.16 \pm 0.02$) > goethite ($0.18 \pm 0.02$) > kaolinite ($0.21 \pm 0.02$) > hematite ($0.28 \pm 0.02$) at 10 mg mL$^{-1}$. An increase in mineral concentration resulted in a significant ($p < 0.05$) decrease in $OD_{600}$. However, in presence of montmorillonite the $OD_{600}$ is stable at about 0.43 for all the mineral concentration studied.

### 3.2. Chemical structure of minerals

To further explore the factors influencing the bacterial growth by montmorillonite, electron paramagnetic resonance (EPR) spectra were used. The EPR spectra revealed that both the kaolinite and montmorillonite samples were dominated by signals from structural Fe(III), which were located around 1600 gauss (g ~ 4.3) (Fig. 2). Iron oxides, which are commonly associated with these minerals produce a broad signal

centred on ~ 3500 gauss (g ~ 2.0). However, the relatively weak resonance indicated

that neither sample had appreciable amounts of iron oxides associated with it. The

montmorillonite also showed a signal from Mn(II) and a free radical, whereas the free

radical signal in the kaolinite was very weak, and there was no evidence of any Mn(II)

signal in this sample.

### 3.3. Generation of HO$^\bullet$

A 12 h cultivation of J12 in the presence of different minerals revealed that

generation of HO$^\bullet$ radicals in the cases of montmorillonite, kaolinite and hematite

was similar ($p > 0.05$) to the control at low concentration (i.e., 5 mg mL$^{-1}$) but

significant different ($p < 0.05$) at high concentration (i.e., 25 mg mL$^{-1}$) (Fig. 3).

However, presence of goethite and ferrihydrite significantly increased the production

of HO$^\bullet$ radicals, which increased with an increase in their concentration. Specifically,

in ferrihydrite treatments, the concentration of HO$^\bullet$ was approximately 260 nM at 5

and 10 mg mL$^{-1}$ but increased significantly to 450 nM at 25 mg mL$^{-1}$. In addition,

the generation of HO$^\bullet$ at early growth (i.e., 2 h) was only detected with ferrihydrite at

both 10 and 25 mg mL$^{-1}$ and with goethite at 25 mg mL$^{-1}$ (Fig. S6).

### 3.4. Iron chemistry and its correlation with HO$^\bullet$ and OD$_{600}$

To explore the factors affecting the generation of HO$^\bullet$ and the inhibition of J12, we

examined iron chemistry and its correlation with HO$^\bullet$ and OD$_{600}$ (Fig. 5). Much

more soluble Fe at 12 h was released from Fe(III)-containing minerals (6.7-27, 21-36

and 41-107 mg L$^{-1}$ for hematite, goethite and ferrihydrite, respectively) than from

montmorillonite (~0.3 mg L$^{-1}$), kaolinite (~0.6 mg L$^{-1}$), and control (~0.4 mg L$^{-1}$)

(Fig. 5a). With the increase of concentration, soluble Fe significantly ($p < 0.05$) increased at both 2 h and 12 h for ferrihydrite, only at 12 h for goethite. As for hematite, significant ($p < 0.05$) increase was only observed from 5 to 10 mg $L^{-1}$ at 12 h (Fig. S7). The solubility of Fe was closely related to redox potential and pH value (Fig. S8). Results showed that Eh of bacteria-mineral mixture after incubation was generally lower than the suspension of minerals alone (Table S5), suggesting that the redox potential was decreased by the interaction between mineral and J12. Furthermore, the solution pH was determined after 12 h growth of J12 with different minerals and with no minerals (control) (Fig. 4). The range of solution pH varied from 4 to 6 for all of the treatments, except for ferrihydrite treatment with a pH near 7. For all of the examined minerals, the trends at 12 h were similar in the following order (total Fe and Fe(II)): ferrihydrite (760-3588 and 182-488 mg $L^{-1}$) > goethite (48-127 and 31-94 mg $L^{-1}$) > hematite (15-82 and 9-35 mg $L^{-1}$) > montmorillonite (5-10 and 4-8 mg $L^{-1}$), kaolinite (10-12 and 4-9 mg $L^{-1}$) or control (7 and 6 mg $L^{-1}$) (Fig. 5b-5c). A significant difference of total Fe in solutions containing 25 mg $mL^{-1}$ ferrihydrite between 2 h and 12 h may be attributable to the aging of a portion of ferrihydrite to its more crystalline counterparts, as revealed by μ-XRF, which could not be dissolved by the modified 1,10-phenanthroline method.

Furthermore, a positive correlation exists between HO$^{\bullet}$ and soluble Fe content ($R = 0.92$, t = -3.49, $p = 0.003$) and Fe(II) ($R = 0.98$, t = -4.28, $p = 0.001$) (Fig. 5d and 5f, Table S2). However, a significant but negative correlation between OD$_{600}$ and soluble Fe ($R = -0.57$, t = 2.99, $p = 0.009$), and Fe(II) ($R = -0.81$, t = 2.23, $p = 0.038$) was found (Fig. 5g and 5i). Moreover, the correlation between HO$^{\bullet}$ and Fe(III) ($R = 0.94$, t = 1.38, $p = 0.19$), and between OD$_{600}$ and Fe(III) ($R = -0.80$, t = 1.67, $p = 0.116$) were not significant (Fig. 5e and 5h). To test whether the release of Fe(III) to solution

inhibit the growth of J12 via a free-radical mechanism, we replaced Fe(III)-containing

minerals by adding a series of concentrations of $Fe(NO_3)_3$, i.e., 0, 50 and 100 mg $L^{-1}$,

in the cultivation experiments with the final pH of 7.2. The results showed that

addition of Fe(III) can inhibit the growth of J12 (25-50%) by producing an additional

$HO^\bullet$ concentration of 15 nM (Fig. S9), supporting the role of Fe(III) ion from solution

in the initialization of a free-radical reaction. In addition, the inhibition of soluble Fe

on J12 was more important in the concentration of 100 mg $L^{-1}$ than that of 50 mg $L^{-1}$

while $HO^\bullet$ production still kept the same between those two concentrations (Fig. S9).

The reason of this phenomenon may attributable to the intracellular oxidative damage

of soluble Fe that penetrated into cells and triggering of intracellular ROS generation.

In addition, we also examined soluble Al during the cultivation experiments (Fig.

6a) and found a high concentration of Al in the montmorillonite and kaolinite

solutions. However, almost no correlation was found between soluble Al and $HO^\bullet$ ($R =$

-0.35, t = -3.36, $p = 0.004$) and $OD_{600}$ ($R = 0.30$, t = 2.24, $p = 0.041$) (Fig. 6b-6c).

**3.5. In situ observation of Fe species and the distribution of organic functional**

**groups**

To explore the critical role of Fe chemistry in the inhibition of the growth of J12, we

used correlative μ-XRF and SR-FTIR analyses for *in situ* measurement of the

distribution of Fe species and EPS on the surface of ferrihydrite. The μ-XRF

spectromicroscopy showed a distinct density of Fe distributed on iron particles (Fig.

7a). Two positions were selected for identifying the coordination state and species of

Fe by μ-XANES spectra. Using hematite, goethite, ferrihydrite, iron(II) oxalate, and

iron(III) oxalate as reference compounds, the linear combination fitting (LCF) results

from Fe K-edge μ-XANES spectra indicated that ferrihydrite was dominant (~82%),

with a lesser percentage (~17%) of $FeC_2O_4$ among the mineral particles (Spot A in Fig. 7b and Table S3). However, considerable percentages of hematite (~13%), goethite (~19%) and $FeC_2O_4$ (~25.9%) were present on the edge of these mineral particles (Spot B in Fig. 7b and Table S3).

Furthermore, the SR-FTIR spectromicroscopy (Fig. 7c) showed that ferrihydrite (Fe-OH, 3344 cm$^{-1}$) had a similar distribution pattern with lipid (C-H, 2921 cm$^{-1}$), amide I (C=O, 1632 cm$^{-1}$), and amide II (C-N, 1513 cm$^{-1}$). However, polysaccharides (C-OH, 1030 cm$^{-1}$) seemed to be distributed only in the big mineral particles. Furthermore, correlation analysis confirmed significant ($R \geq 0.68$, $p < 0.003$) linear correlations between ferrihydrite and these EPS (i.e., lipid, amide I, amide II, and polysaccharides) (Fig. S10).

### 3.6. Effect of the presence of J12 on surface Fe species

XPS analysis was conducted to investigate the oxidation state of Fe in the interface between iron minerals and J12 (Fig. 8). The shift of the Fe $2p_{3/2}$ peak of 0.5 eV was observed between raw ferrihydrite and ferrihydrite after 12 h of cultivation with bacteria (Fig. 8a). Four Fe $2p_{3/2}$ peaks at 709.5 eV, 710.3 eV, 711.5 eV, 713.1 eV appeared in the F + bacteria treatment (Fig. 8b-8c). The peaks at 710.3 eV, 711.5 eV and 713.1 eV are regarded as multiplet peaks of Fe(III), but the peak at 709.5 eV is interpreted as Fe(II) (Grosvenor et al., 2004). Interestingly, the area of the peak at 709.5 eV was bigger in the F + bacteria treatment than that in F - bacteria treatment (Fig. 8b-8c). Based on the reaction 1, $HO_2^•$ should be the oxidant products.

### 4. Discussion

## 4.1. Effect of Al(III)-containing minerals on the inhibition of J12 growth

Our results showed that kaolinite (1 : 1 layer-type) resulted in significant inhibition of the growth of J12, but montmorillonite (2 : 1 layer-type) remarkably accelerated its growth (Fig. 1). Similarly, recent studies have shown the toxic effects of aluminosilicate on microorganisms (Liu et al., 2016; Wilson and Butterfield, 2014), but the bacterial activity was not inhibited by the interfacial interactions between montmorillonite and bacteria (Wilson and Butterfield, 2014). It should be noted that the presence of minerals may potentially interfere with the measurement of cell numbers in Fig. 1. In this study, we subsampled the experimental cultures and diluted them in fresh medium so that both clay particles and J12 were 200× less concentrated (Fig. S3), following the protocol of McMahon et al. (2016). As a result, the effect of mineral concentration may be minimal. In addition, plating the bacteria by evaluating populations by counting colonies may act as a complementary method for $OD_{600}$ and needs to be investigated in the future. Furthermore, the association of a cell labeling with DAPI and a count of labeled cells with flow cytometry (or fluorescence microscopy) is also an alternative choose.

It is generally accepted that diverse bacteria are susceptible to Al(III). In the present study, the amount of aqueous Al(III) exceeded 2 mg $L^{-1}$ for all kaolinite experiments while its concentration was negligible in the presence of montmorillonite during the early growth of J12 (Fig. 6). It is worth noting that >2 mg $L^{-1}$ of aqueous Al(III) was detected for montmorillonite experiments with the passage of time (Fig. 6); however, the growth of J12 was not inhibited (Fig. 1). This may be attributed to the adsorption of aqueous Al(III) by bacterial EPS, which further protected bacteria from damage (Wu et al., 2014). However, direct evidence is lacked in this study and thus

further investigation is needed to address this issue. Thus, the inhibition of bacterial activity by kaolinite may possibly be attributed to the toxicity of aqueous Al(III). Specifically, Al(III) reacts with membrane phospholipids and then increases membrane permeability that leads to the inactivation of bacteria (Londono et al., 2017).

In addition, some essential elements (e.g., Mg and P) can be affected by Al(III) for bacterial absorption, which could also limit bacterial growth (Piña and Cervantes, 1996; Londono et al., 2017). Furthermore, the formation of some Al intermediates by the decreasing pH, such as $Al_{13}O_4(OH)_{24}{}^{7+}$, is also suggested to be more toxic for bacterial growth (Amonette et al., 2003; Liu et al., 2016). However, we did not detect

a significant decrease of pH in this study (Fig. 4), suggesting that the formation of some Al intermediates may be slightly.

## 4.2. Inhibition of J12 by Fe(III)-containing minerals via a free-radical mechanism

Our results showed that Fe(III)-containing minerals resulted in higher generation of

390 $HO^{\bullet}$ and had higher inhibition efficiency on J12 than Al(III)-containing minerals (Figs. 1 and 3). Fe is widely known as a transition metal that might cause microbial inactivation through ROS-mediated cellular damage, i.e., genotoxicity, protein dysfunction and impaired membrane function (Lemire et al., 2013). Inhibition of heterotrophic bacteria by Fe minerals is generally attributed to the generation of $HO^{\bullet}$

through a Fenton reaction (Morrison et al., 2016) or Fenton-like reaction (Garrido-Ramírez et al., 2010). Due to its amorphous structure, high reactive surface area and solubility, ferrihydrite ($\sim$200-300 $m^2 \ g^{-1}$) is more likely to physically interact with bacterial surfaces than hematite ($\sim$30 $m^2 \ g^{-1}$) and goethite ($\sim$20 $m^2 \ g^{-1}$)

(Schwertmann and Cornell, 2007; Lemire et al., 2013). A recent study demonstrated

that metal oxide nanoparticles produced more ROS than bulk metal oxides (Wang et

al., 2017a). In this study, we observed higher $HO^•$ formation and stronger inhibition of

J12 in ferrihydrite treatments (Figs. 1 and 3), suggesting that reactive surface area and

solubility has a significant effect on enhancing formation of $HO^•$ and inhibition

activity of J12. Reactive mineral surfaces can catalyze $HO^•$ generation or act as

"carriers" where $HO^•$-inducing materials are adsorbed (Schoonen et al., 2006). In our

experiment, there was a lesser amount of $HO^•$ produced with the different

concentrations of aqueous $Fe(NO_3)_3$ (Fig. S9) than with the iron minerals (Fig. 3),

which was in line with other studies (Kwan and Voelker, 2003; Wang et al., 2017b).

Therefore, we deduced that $HO^•$ may mainly generate on the mineral surface, partly

due to the positive charge of mineral surface (Tombácz and Szekeres, 2006) but the

negative charge of microbes (Jucket et al., 1996).

A recent study demonstrated that surface rather than aqueous Fe(II) plays a

dominant role in producing extracellular $HO^•$ that damage cell membrane lipid

revealed by in-situ imaging (Wang et al., 2017b). The following reactions (equations

3-4) reveal that the generation of $HO^•$ is catalyzed by surface Fe(II) (Kwan and

Voelker, 2003; Polerecky et al., 2012):

$$\equiv Fe(III) + H_2O_2 \rightarrow \equiv Fe(II) + H^+ + HO_2^• \tag{3}$$

$$\equiv Fe(II) + H_2O_2 \rightarrow \equiv Fe(III) + HO^• + OH^- \tag{4}$$

In this study, a substantial amount of Fe(II) was generated by ferrihydrite,

approximately 4-fold higher than soluble Fe (Fig. 5). This amount of Fe(II) included

two portions: one existed in solution, another was derived from the mineral surface.

To further confirm the generation of surface Fe(II), Fe K-edge μ-XANES analysis were used, and they showed that ferrihydrite presented various Fe species, and Fe(II) increased from 17.3% among the mineral particles (A position) to 25.9% in the edge of mineral particles (B position) (Fig. 7 and Table S3). High percentage of the less stable ferrihydrite (Table S3) may be attributable to the stabilization role of produced EPS (Fig. 7c) by J12 to minerals. It is consistent with a previous finding in the cultivation of fungi with minerals (Li et al., 2016). The stabilization role of EPS on meta-stable ferrihydrite was mainly identified as its combination into the network structure of minerals, which prevents the formation of crystalline minerals (Braunschweig et al., 2013). Note that the LCF results are dependent on the range of compounds used to generate the reference spectrum library, which is one drawback of LCF. To further support the LCF results, XPS being a near-surface sensitive technique is also used to detect the production of ferrous iron at the surface of the iron oxides, owing to a greater certainty than LCF and XANES to demonstrate the presence of ferrous iron by fitting multiplet-splitting models (Grosvenor et al., 2004). According to the XPS analysis, the Fe $2p_{3/2}$ peak shifted from high energy (F – bacteria) to low energy (F + bacteria) (Fig. 8), revealing that Fe(II) was produced on the surface of ferrihydrite during cultivation.

In addition to Fenton-like reactions (Garrido-Ramírez et al., 2010), Fe(II) can also be generated by catalyzing a series of intracellular reductants (e.g., glutathione, NAD(P)H, L-cysteine and FADH$_2$) (Imlay, 2003). Other metabolically formed oxidants released by bacteria may also contribute to Fe(II) oxidation (Melton et al., 2014). Subsequently, the oxidation of Fe(II) to Fe(III) is followed by a reduction of the Fe(III) to Fe(II) (Melton et al., 2014). In addition, many microorganisms are thought to transfer electrons between their cytoplasmic membranes and extracellular

minerals through a network of redox and structural c-type cytochromes (c-Cyts) and flavins (Shi et al., 2016). The redox cycling of Fe during interfacial interactions between Fe(III)-containing minerals and bacteria accelerates the generation of HO$^{\bullet}$ (Page et al., 2013).

The responses of the inhibition activity of J12 followed the order Fe(III)-containing minerals > Fe(NO$_3$)$_3$ > Control (Figs. 1 and S9). Intracellular oxidative toxicity also caused by soluble Fe(III) played an important role in the inhibition activity (Schoonen et al., 2006). We deduced that inhibition of J12 with Fe(III)-containing minerals mainly depends on the coupled effect of soluble Fe, surface Fe(II), and extracellular HO$^{\bullet}$.

## 4.3. Inhibition of J12 growth by a free-radical mechanism and its implications for soil carbon storage

In this study, we proposed a schematic of Fe(III)-containing minerals inhibiting J12 growth through a free-radical mechanism (Fig. 9). Surface Fe(II) is produced from the reduction of Fe(III) on the surface of Fe(III)-containing minerals, promoting the production of extracellular HO$^{\bullet}$ through the Fenton or Fenton-like reactions (Garrido-Ramírez et al., 2010). Soluble Fe(II) and Fe(III) released from minerals can penetrate into the cell membranes, thereby inducing intracellular oxidative damage (Williams et al., 2011). Oxidative damage of HO$^{\bullet}$ may induce the damage of a membrane lipid and cardiolipin that can lead to heterotrophic bacterial inactivation (Wang et al., 2017). In soil, heterotrophic bacteria are the main driver of soil carbon decomposition and greenhouse gas emission. As a result, the inactivation of heterotrophic bacteria results in protection of carbon from microbial degradation.

Except for decomposition of soil organic carbon (SOC), the presence of HO$^{\bullet}$ can also stabilize C in soil via a rapid formation of new intermolecular covalent bonds among soil components (Piccolo et al., 2011). Formation of new intermolecular covalent bonds increases the recalcitrance of SOC. In addition, the generation of free radicals may also have indirect effects on J12 growth via substrate availability (Table S4).

Substrate availability is improved in the presence of radicals, owing to the depolymerization role of radicals on the complex substrates.

  Microbes affect the cycling of SOC, and their products are important components of SOC (Kögel-Knabner, 2002; Kleber and Johnson, 2010; Schmidt et al., 2011; Liang et al., 2017). The mobilized Fe can be easily transformed into the newly-formed

reactive Fe (hydro)oxides (especially poorly crystalline Fe oxides) (Kleber et al., 2005; Yu et al., 2017), which will promote the formation of organo-mineral associations that are chemically more stable (Koegel-Knabner et al., 2008). In this study, we suggest that soil carbon cycle is partly regulated by Fe minerals (i) by the formation of organo-mineral complexes (Kögel-Knabner, 2002; Kleber and Johnson, 2010;

Schmidt et al., 2011) and (ii) by the bacterial development inhibition. However, it should be noted that NB medium containing casein and meat hydrolysates is only a medium that enables the growth of J12 in this study, and it is far away from organic matter decomposition or substrates available in soil systems. Further investigation should be conducted to explore the effect of microbe-driven Fenton-like reaction on

the storage of SOC in soil system in the future.

## 5. Conclusions

Kaolinite, hematite, goethite and ferrihydrite had a significant inhibitory effect on the growth of *Pseudomonas brassicacearum* J12, which was more prominent with a higher concentration, following the order 25 mg mL$^{-1}$ > 10 mg mL$^{-1}$ > 5 mg mL$^{-1}$. In contrast, montmorillonite promoted the growth of J12, which was independent on its concentration. Compared to Al(III)-containing minerals, Fe(III)-containing minerals promoted more HO$^{\bullet}$ generation and thus increased suppression to J12 via a free-radical mechanism. Furthermore, our results revealed that surface Fe(II) was produced on the mineral surface that may act as a catalyst, promoting the generation of HO$^{\bullet}$ rather than soluble Fe. The generation of HO$^{\bullet}$ by Fe(III)-containing minerals follows the order ferrihydrite > goethite > hematite. In summary, our findings indicate that the inhibition of heterotrophic bacteria with Fe(III)-containing minerals mainly depends on the coupled effect of soluble Fe and extracellular HO$^{\bullet}$, which may further contribute to soil carbon storage.

The Supplement related to this article is available online at **http://www.biogeosciences.net/.**

*Acknowledgments.* We thank the staff for their support at BL01B beamline of National Center for Protein Sciences Shanghai (NCPSS) and BL15U1 at Shanghai Synchrotron Radiation Facility (SSRF), for assistance during data collection. This work was funded by the National Key Research and Development Program of China (2017YFD0800803) and the National Natural Science Foundation of China (41671294 and 41371248).

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

**Figure Captions**

**Figure 1.** Optical density at 600 nm ($OD_{600}$) of 8-h-old *Pseudomonas brassicacearum*

J12 subcultures taken after 12 h growth with different minerals and with no minerals

(control). Al-containing minerals: K, kaolinite; M, montmorillonite. Fe-containing

minerals: H, hematite; G, goethite; F, ferrihydrite. C, Control (i.e., no mineral). Gray,

magenta and cyan represent the mineral concentration of 5, 10 and 25 mg mL$^{-1}$,

respectively. Values are the mean ± SE (n = 3).

**Figure 2**. Wide scan EPR spectra of both the kaolinite and montmorillonite.

**Figure 3.** Generation of hydroxyl radical (HO$^{\bullet}$) after 12 h growth of *Pseudomonas*

*brassicacearum* J12 with different minerals and with no minerals (control).

Al-containing minerals: K, kaolinite; M, montmorillonite. Fe-containing minerals: H,

hematite; G, goethite; F, ferrihydrite. C, Control (i.e., no mineral). Gray, magenta and

cyan represent the mineral concentration of 5, 10 and 25 mg mL$^{-1}$, respectively.

Values are the mean ± SE (n = 3).

**Figure 4.** Determination of pH after 12 h growth of *Pseudomonas brassicacearum*

J12 with different minerals and with no minerals (control). Al-containing minerals: K,

kaolinite; M, montmorillonite. Fe-containing minerals: H, hematite; G, goethite; F,

ferrihydrite. C, Control (i.e., no mineral). Gray, magenta and cyan represent the

mineral concentration of 5, 10 and 25 mg mL$^{-1}$, respectively. Values are the mean ±

SE (n = 3).

**Figure 5.** Iron chemistry (a-c) and its correlation with hydroxyl radical (HO$^\bullet$) (d-f) as well as optical density at 600 nm (OD$_{600}$) (g-i). (a) soluble Fe. (b) total Fe. (c) Fe(II). (d) soluble Fe vs HO$^\bullet$. (e) total Fe vs HO$^\bullet$. (f) Fe(II) vs HO$^\bullet$. (g) soluble Fe vs OD$_{600}$. (h) total Fe vs OD$_{600}$. (i) Fe(II) vs OD$_{600}$. Al-containing minerals: K, kaolinite; M, montmorillonite. Fe-containing minerals: H, hematite; G, goethite; F, ferrihydrite. C, Control (i.e., no mineral). Values in (a-c) are the mean $\pm$ SE (n = 3).

**Figure 6.** (a) Production of soluble Al after 2 h and 12 h cultivation. Al-containing minerals: K, kaolinite; M, montmorillonite. Fe-containing minerals: H, hematite; G, goethite; F, ferrihydrite. C, control, no minerals. Values in (a) are the mean $\pm$ SE (n = 3). Correlation analysis between HO$^\bullet$ concentration and soluble Al is shown in (b), and correlation analysis between OD$_{600}$ and soluble Al is shown in (c).

**Figure 7.** Correlative micro X-ray fluorescence ($\mu$-XRF) and synchrotron-based Fourier transform infrared (SR-FTIR) analysis of the thin section from the cultures of the 25 mg/mL ferrihydrite treatment after 12 h cultivation. (a) $\mu$-XRF map. (b) The LCF fitting of $\mu$-X-ray absorption near-edge structure (XANES) analysis the selected regions of interest (ROI) region (i.e., A and B). (c) SR-FTIR maps. Red color in (a) represents high density of Fe, followed by orange, yellow, green, little green, and purple.The color scale in (c) is a relative scale for each peak height and does not allow quantitative comparisons between peaks.

**Figure 8.** (a) Fe 2p X-ray photoelectron spectroscopy (XPS) spectra of ferrihydrite samples, F+bacteria and F-bacteria; (b-c) Fe 2p 3/2 spectra of F+bacteria and

F-bacteria, respectively, during the cultivation (12 h). F+bacteria, ferrihydrite with bacteria; F-bacteria, ferrihydrite without bacteria. In subfigure (b) and (c), dark, orange and other lines represents the raw spectrum, fitted spectrum and the component of fitted Fe species.

**Figure 9.** Schematic of the inhibition of heterotrophic bacteria by Fe(III)-containing minerals through a free-radical mechanism. ①-④ represent the processes occurring at heterotrophic bacterial-mineral interfaces and are detailed in the main text. ① Production of HO$^•$ through the Fenton or Fenton-like reactions; ② Direct inhibation of heterotrophic bacteria by HO$^•$; ③ Indirect inhibition of heterotrophic bacteria by HO$^•$; ④ Intracellular inhibition of heterotrophic bacteria by soluble Fe.

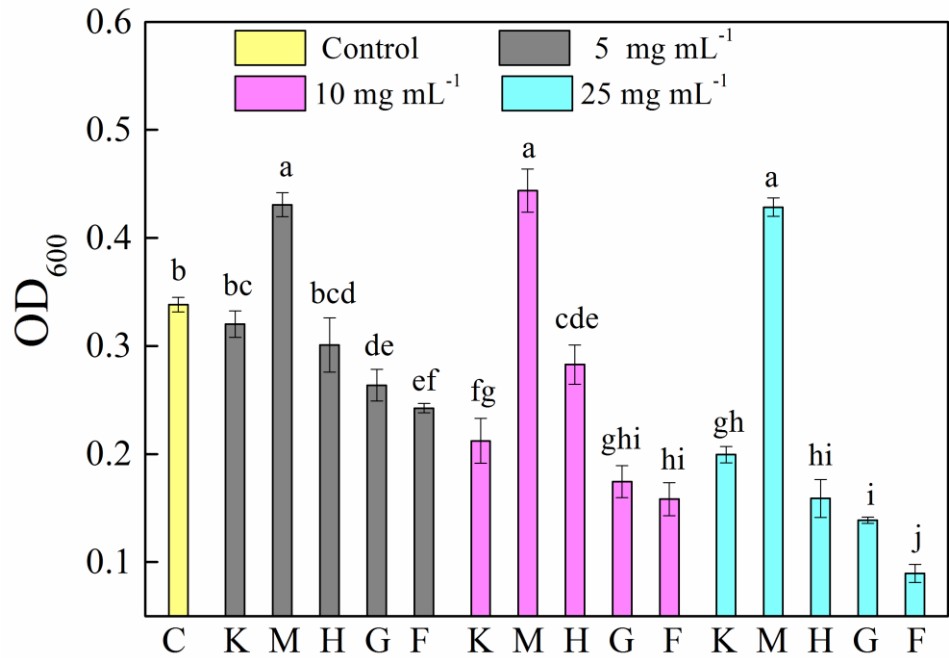

**Figure 1.** Optical density at 600 nm ($OD_{600}$) of 8-h-old *Pseudomonas brassicacearum* J12 subcultures taken after 12 h growth with different minerals and with no minerals (control). Al-containing minerals: K, kaolinite; M, montmorillonite. Fe-containing minerals: H, hematite; G, goethite; F, ferrihydrite. C, Control (i.e., no mineral). Gray, 740 magenta and cyan represent the mineral concentration of 5, 10 and 25 mg mL$^{-1}$, respectively. Values are the mean ± SE (n = 3).

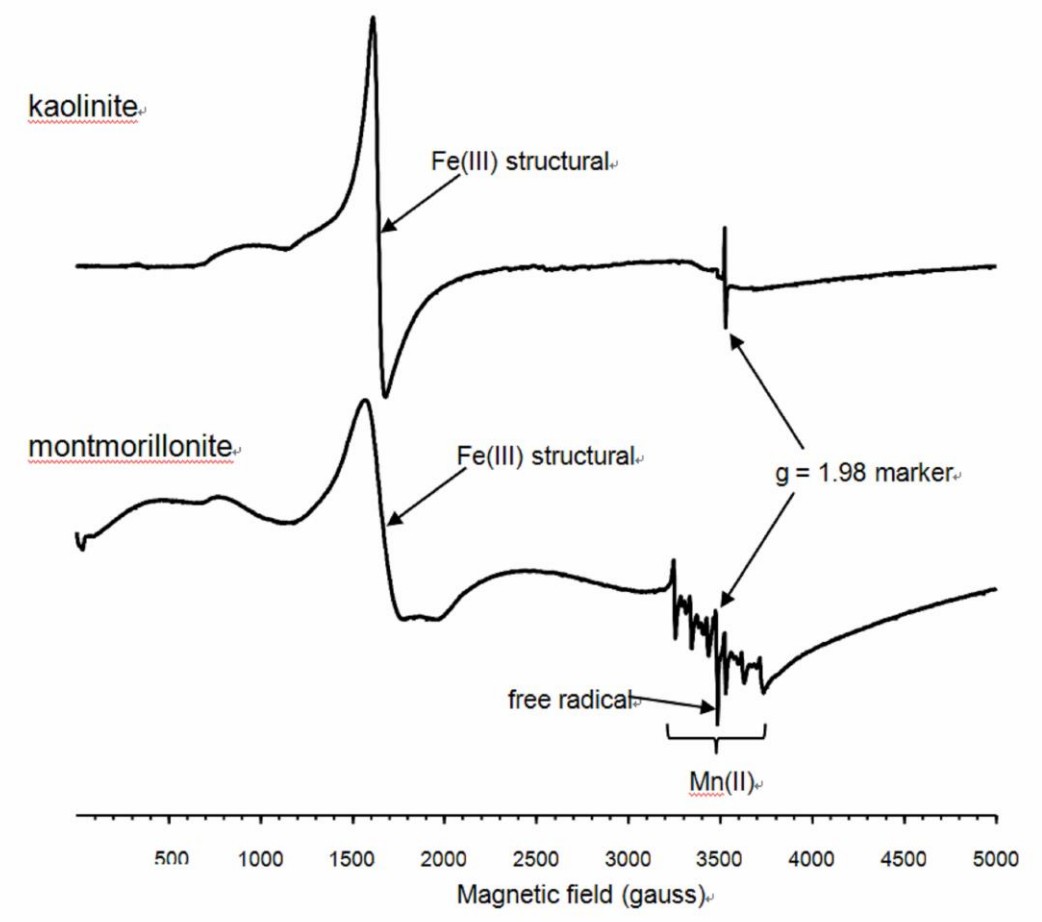

**Figure 2**. Wide scan EPR spectra of both the kaolinite and montmorillonite.

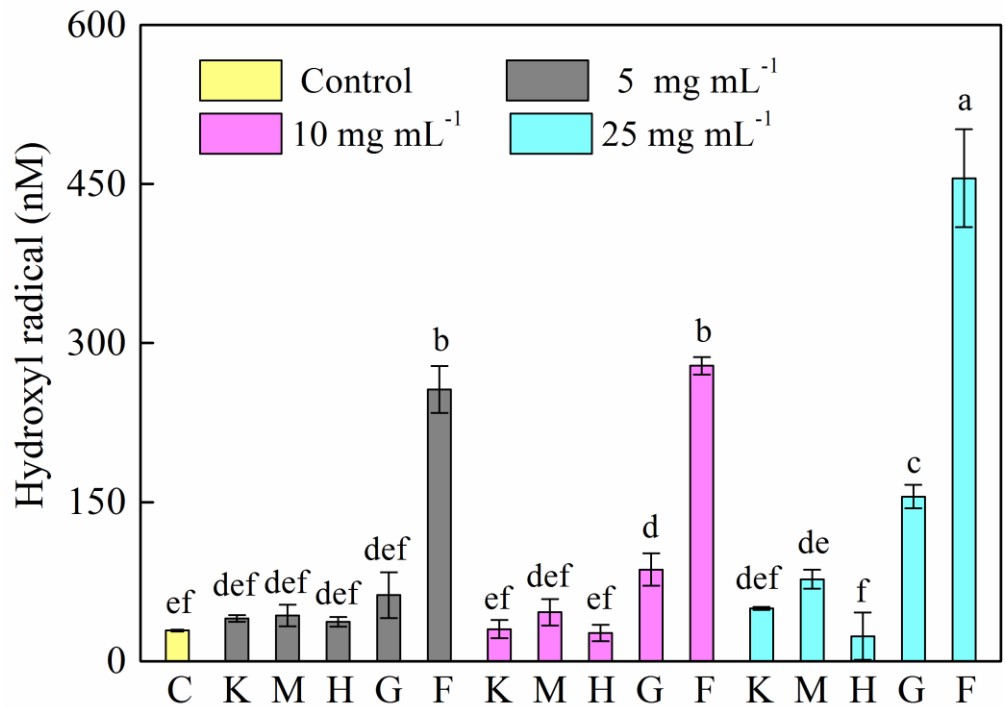

**Figure 3.** Generation of hydroxyl radical (HO˙) after 12 h growth of *Pseudomonas*

*brassicacearum* J12 with different minerals and with no minerals (control).

Al-containing minerals: K, kaolinite; M, montmorillonite. Fe-containing minerals: H,

hematite; G, goethite; F, ferrihydrite. C, Control (i.e., no mineral). Gray, magenta and

cyan represent the mineral concentration of 5, 10 and 25 mg mL$^{-1}$, respectively.

Values are the mean $\pm$ SE (n = 3).

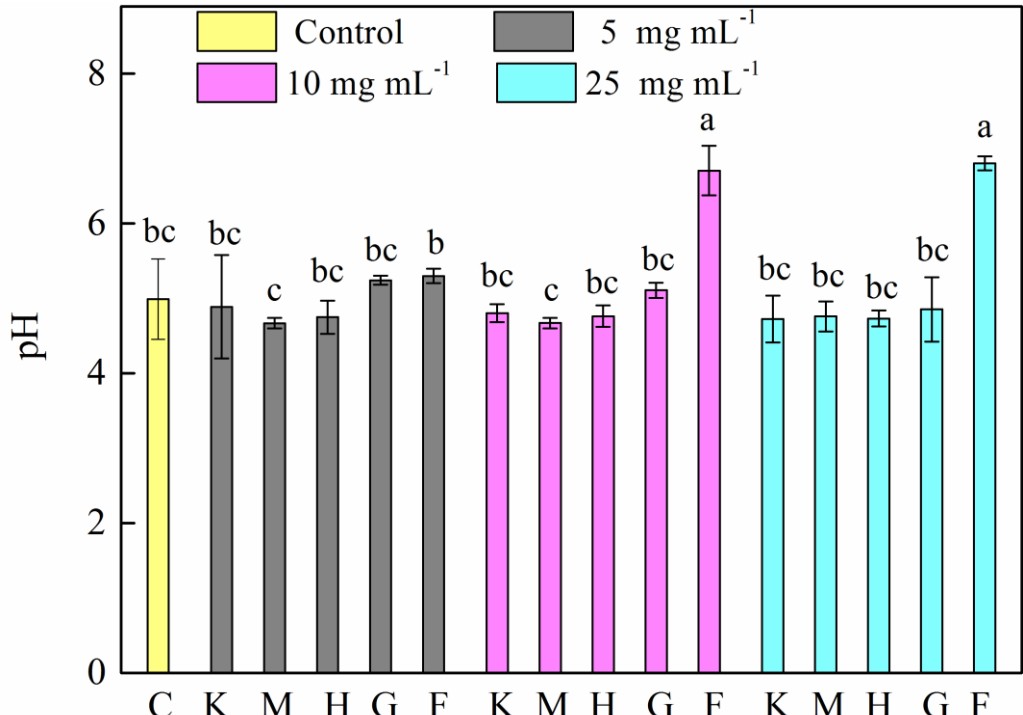

**Figure 4.** Determination of pH after 12 h growth of *Pseudomonas brassicacearum* J12 with different minerals and with no minerals (control). Al-containing minerals: K, kaolinite; M, montmorillonite. Fe-containing minerals: H, hematite; G, goethite; F, ferrihydrite. C, Control (i.e., no mineral). Gray, magenta and cyan represent the mineral concentration of 5, 10 and 25 mg mL$^{-1}$, respectively. Values are the mean $\pm$ SE (n = 3).

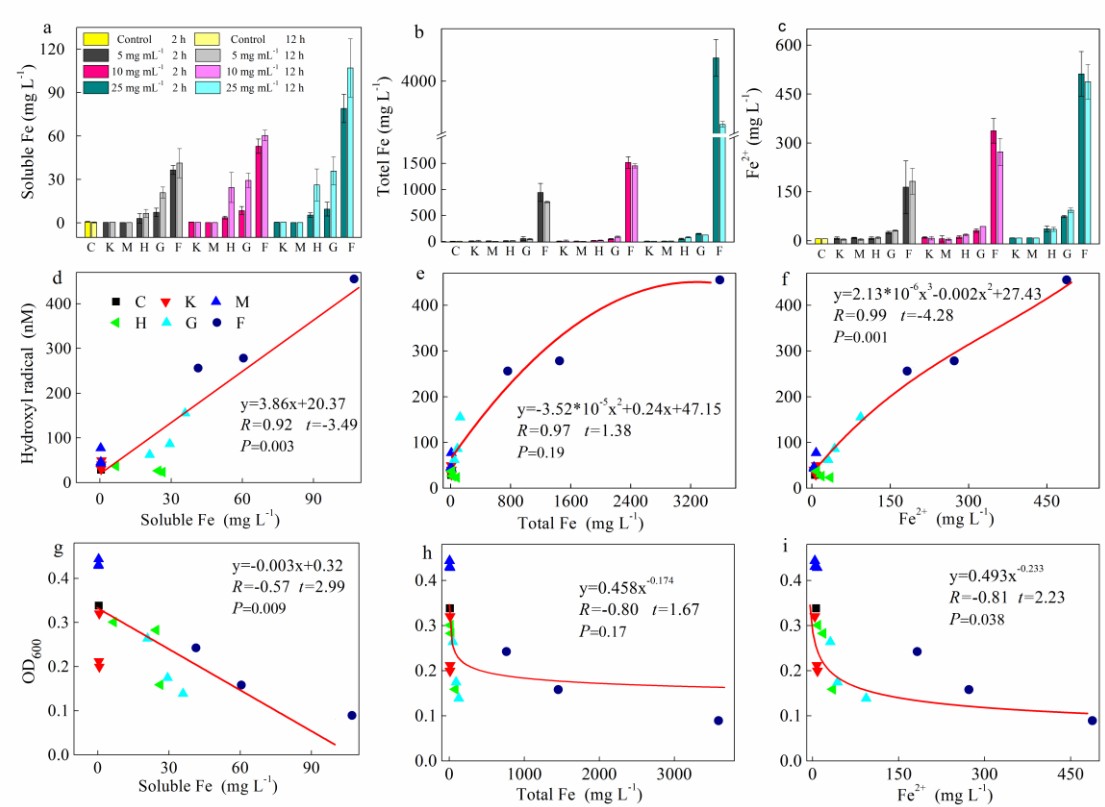

**Figure 5.** Iron chemistry (a-c) and its correlation with hydroxyl radical (HO•) (d-f) as well as optical density at 600 nm ($OD_{600}$) (g-i). (a) soluble Fe. (b) total Fe. (c) Fe(II). (d) soluble Fe vs HO•. (e) total Fe vs HO•. (f) Fe(II) vs HO•. (g) soluble Fe vs $OD_{600}$. (h) total Fe vs $OD_{600}$. (i) Fe(II) vs $OD_{600}$. Al-containing minerals: K, kaolinite; M, montmorillonite. Fe-containing minerals: H, hematite; G, goethite; F, ferrihydrite. C, Control (i.e., no mineral). Values in (a-c) are the mean ± SE (n = 3).

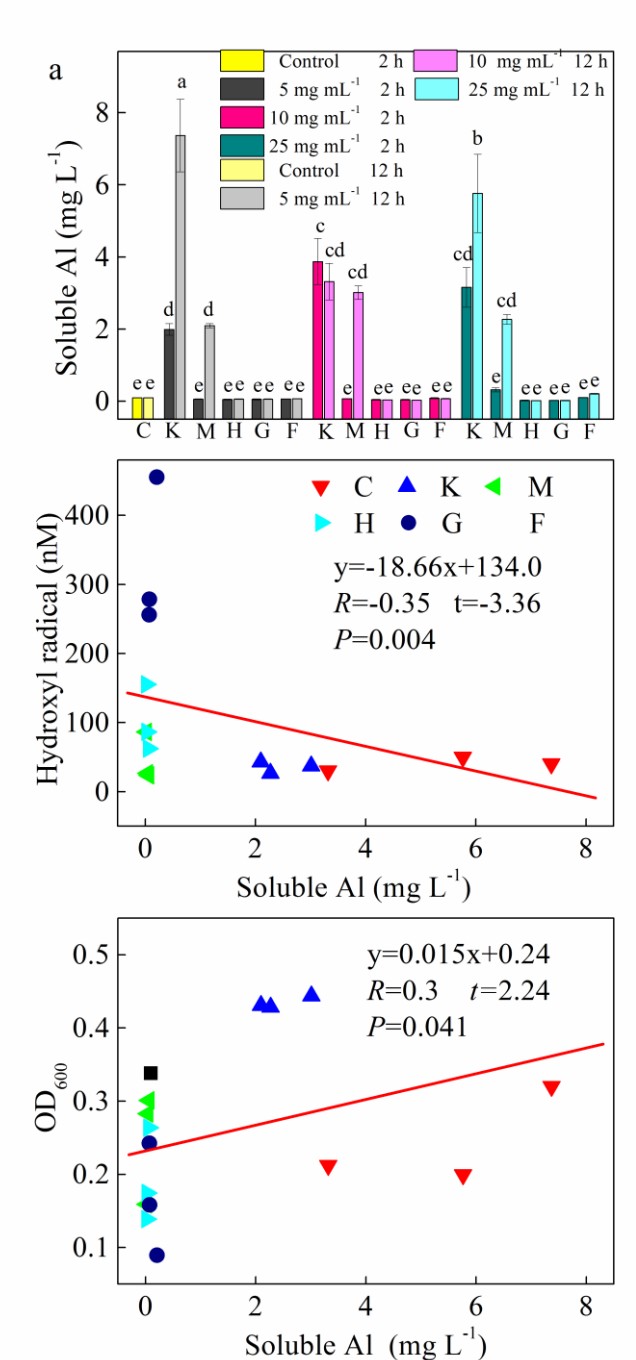

**Figure 6.** (a) Production of soluble Al after 2 h and 12 h cultivation. Al-containing minerals: K, kaolinite; M, montmorillonite. Fe-containing minerals: H, hematite; G, goethite; F, ferrihydrite. C, control, no minerals. Values in (a) are the mean ± SE (n = 3). Correlation analysis between HO• concentration and soluble Al is shown in (b), and correlation analysis between $OD_{600}$ and soluble Al is shown in (c).

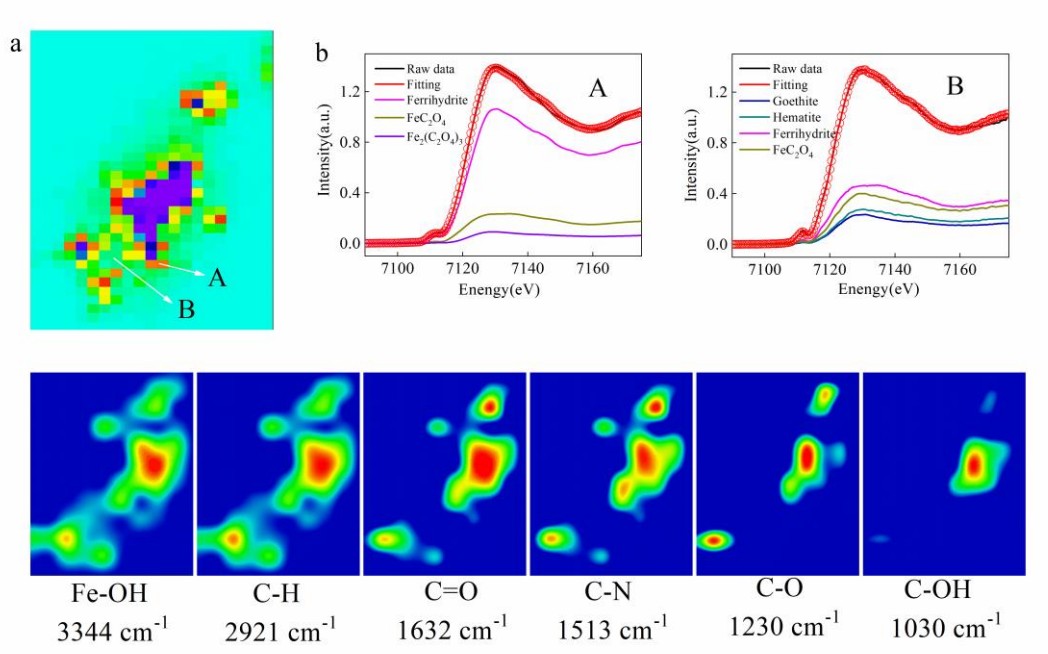

**Figure 7.** Correlative micro X-ray fluorescence (μ-XRF) and synchrotron-based Fourier transform infrared (SR-FTIR) analysis of the thin section from the cultures of the 25 mg/mL ferrihydrite treatment after 12 h cultivation. (a) μ-XRF map. (b) The LCF fitting of μ-X-ray absorption near-edge structure (XANES) analysis the selected regions of interest (ROI) region (i.e., A and B). (c) SR-FTIR maps. Red color in (a) represents high density of Fe, followed by orange, yellow, green, little green, and purple. Red color in (c) indicates the highest intensity of functional groups, followed by yellow, green, and blue. The color scale in (c) is a relative scale for each peak height and does not allow quantitative comparisons between peaks.

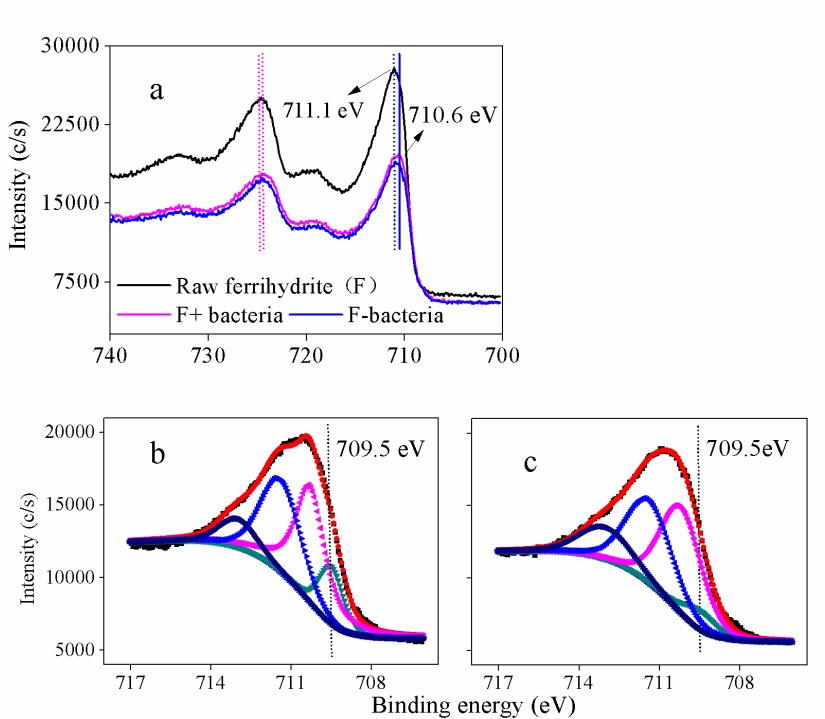

**Figure 8.** (a) Fe 2p X-ray photoelectron spectroscopy (XPS) spectra of ferrihydrite samples, F+bacteria and F-bacteria; (b-c) Fe 2p 3/2 spectra of F+bacteria and F-bacteria, respectively, during the cultivation (12 h). F+bacteria, ferrihydrite with bacteria; F-bacteria, ferrihydrite without bacteria. In subfigure (b) and (c), dark, orange and other lines represents the raw spectrum, fitted spectrum and the component of fitted Fe species.

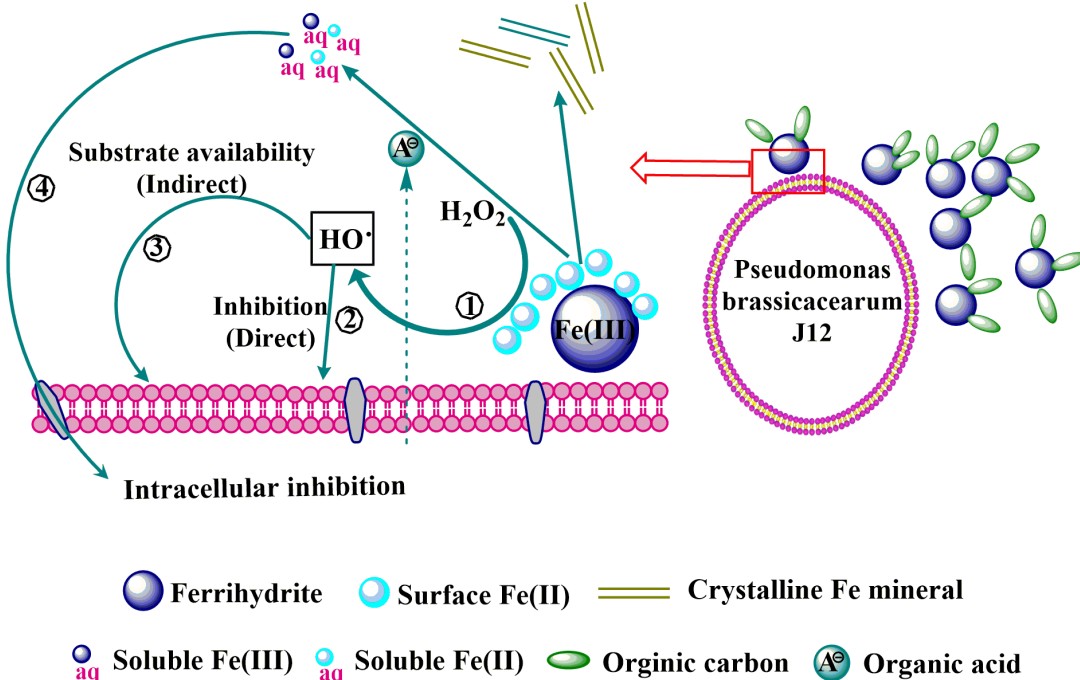

**Figure 9.** Schematic of the inhibition of heterotrophic bacteria by Fe(III)-containing minerals through a free-radical mechanism. ①-④ represent the processes occurring at heterotrophic bacteria-mineral interfaces and are detailed in the main text. ① Production of HO$^{\bullet}$ through the Fenton or Fenton-like reactions; ② Direct inhibation of heterotrophic bacteria by HO$^{\bullet}$; ③ Indirect inhibition of heterotrophic bacteria by HO$^{\bullet}$; ④ Intracellular inhibition of heterotrophic bacteria by soluble Fe.