# Peer review of "Iron minerals inhibit the growth of *Pseudomonas brassicacearum*J12 via a free-radical mechanism: Implications for soil carbon storage"

_Biogeosciences, 2018_

## Referee Comment (RC1) · Anonymous Referee #1 · 30 Dec 2018

General Comments: This study is a nice contribution to the analysis of mineral-microbe interactions in soils, and it presents some new evidence for the role of iron in diminishing certain bacterial populations. Of particular interest to me was the X-ray photoelectron spectroscopy data demonstrating the presence of Fe(II) on the ferrihydrite surface, suggesting that the microbe (J12) reduces mineral bound Fe(III). This is important, because Fe(II) is then available for Fenton reactions that can produce radicals that damage cell membranes allowing soluble metals to enter the bacterial cell. The work suggests that iron bearing minerals in soils contribute to the preservation of organic C by limiting the productivity of bacteria that degrade carbon.

Specific Comments: Line 108. Your description of the clay minerals used is not adequate for determining the chemical composition and stability of the phyllosilicates. In particular montmorillonite has many chemical components in each crystallographic site that contribute to the mineral surface characteristics (charge distribution) and interlayer cations. Therefore, one must give the chemical formula for the mineral used. One montmorillonite might increase bacterial growth (if it provides nutrients) while another might decrease bacterial growth (if it provides toxins).

Line 128. How were the concentrations of minerals (5, 10, 25 mg/ml) decided? Did you measure the minimum inhibitory and minimum bactericidal concentrations for the J12 bacteria under the pH conditions of the experiment? Were the minerals only hydrated by the growth media? If so, what is the speciation of soluble metals with the components of the media solution?

Line 135. It is unclear why the particle size distribution is presented before and after incubation with bacteria. The increase in particle size after incubation (which would be better demonstrated in a graph than a table) probably results from agglomeration of mineral-bacteria clusters rather than a crystal growth. Is this important to the conclusions? If anything, a measurement of specific surface area of the minerals would be more important to the chemical interactions.

Line 209. For chemical analysis you have filtered the mineral-microbe suspension through 0.45 $\mu$m membrane, which may remove the bacteria, but allows clay size particles through. This is then analyzed by ICP and results reported as 'soluble' Al and Fe. However, the clay particles in this fraction will contribute to the elemental analysis.

Line 225. This analysis presumes that OD600 only reflects absorbance by bacteria, but what is the absorbance of the mineral suspension alone?

Line 245. Mn(II), being redox active, is more likely to produce hydroxyl radical than to scavenge (Zarate-Reyes et al., 2017 Appl Clay Sci; Shi et al, 2016 Nature Sci Rev.)

Line 267. Replace expect with except

Line 270. The production of acids by bacteria is plausible, but the surface area of the minerals is far greater (in general) than that of the bacteria, thus you might consider that each mineral surface attracts or repels H+ and OH- which is the major factor in buffering the fluid pH. You should test this by monitoring the pH (and Eh) of the suspension of minerals alone, compared to the suspension of bacteria alone.

Line 289. This requires H2O2, what is the source of that in the mineral-microbe suspension?

Line 295. Al3+ is not redox active, so why (or how) could it be correlated with production of hydroxyl radical? The toxicity of Al is from interactions with phospholipids, not production of radicals.

Line 305. This analysis is confusing to me. Did you add all of the minerals to this bacterial suspension and you are looking for which mineral dominates?

Line 320. This suggests that the bacteria reduce the mineral Fe, which in turn produces radicals that oxidize the bacteria. Why would bacteria not have a defense against such radicals?

Line 341. A simple measurement of the OD600 on a mineral suspension should answer this definitively.

Line 345. The inhibitory concentration of metals is pH dependent, so unless the work of Illmer and Schinner was at the same pH of your experiment (after 12 hrs) then this reported concentration may not be relevant. You need to determine the MIC and MBC concentrations at the pH of your mineral-microbe mixture (after incubation).

Line 385. The phyllosilicates generally have a negative charge on their more extensive basal surfaces. Positive charges are limited to broken edges of the structure. This may not be true for ferrihydrite and goethite. But remember that hydroxyl radicals only exist for 1ns, so for them to interact with bacteria, there must be an attractions between the

microbe and mineral surface where the radical is generated.

Line 403. It is unclear what you mean by 'stabilization role' here. What does the EPS do to stabilize the ferrihydrite?

Line 412. This is an important result of this study!

Line 434. Do you suggest that the H2O2 required is generated by the bacteria or by dissolved O2 in solutions? Have you monitored the Eh of the solutions?

Supplementary Figures Figure S6. How do you explain the drop in pH from 7.2 even in the control suspension? Figure S8. Why does the control have soluble Al even though there are no minerals in it?

---

## Author Comment (AC1) · 13 Jan 2019

Jan. 13, 2019

Dear Editor,

We now submit the revised manuscript bg-2018-479R1 by Haiyan Du, Guanghui Yu, Fusheng Sun, Muhammad Usman, Bernard A Goodman, Wei Ran, Qirong Shen to Biogeosciences. Our point-by-point responses are detailed below, and all changes are highlighted in the revised manuscript. We are very grateful to Reviewer #1 for great comments, which have truly helped in the improvement of the article. Should you need

to contact me, please use the above email or call me at [86-25-84396221].

Sincerely yours,

Guanghui Yu

College of Resources and Environmental Sciences

Nanjing Agricultural University

Nanjing 210095, P.R. China

Phone: +86-25-84396221

Fax: +86-25-84395212

E-mail: yuguanghui@njau.edu.cn or yuguanghui@tju.edu.cn

Reply to the Comments

Anonymous Referee #1

The authors appreciate the report of Referee #1 and respond as follows.

General Comments: This study is a nice contribution to the analysis of mineral-microbe interactions in soils, and it presents some new evidence for the role of iron in diminishing certain bacterial populations. Of particular interest to me was the X-ray photoelectron spectroscopy data demonstrating the presence of Fe(II) on the ferrihydrite surface, suggesting that the microbe (J12) reduces mineral bound Fe(III). This is important, because Fe(II) is then available for Fenton reactions that can produce radicals that damage cell membranes allowing soluble metals to enter the bacterial cell. The work suggests that iron bearing minerals in soils contribute to the preservation of organic C by limiting the productivity of bacteria that degrade carbon.

Specific Comments:

-Line 108. Your description of the clay minerals used is not adequate for determining

the chemical composition and stability of the phyllosilicates. In particular montmorillonite has many chemical components in each crystallographic site that contribute to the mineral surface characteristics (charge distribution) and interlayer cations. Therefore, one must give the chemical formula for the mineral used. One montmorillonite might increase bacterial growth (if it provides nutrients) while another might decrease bacterial growth (if it provides toxins).

Response (R): In the revised manuscript, the chemical formulas for the clay mineral used were added and also listed as follows.

"Five minerals were selected in this study, including kaolinite (98%, Aladdin Reagent Company, Shanghai, China), montmorillonite (98%, Aladdin Reagent Company, Shanghai, China) and synthetic hematite, goethite and ferrihydrite." (Pages 5-6, Lines 107-109 in the original manuscript)

was changed to

"Five minerals were selected in this study, including kaolinite (Al2O3•2SiO2•2H2O, 98%, Aladdin Reagent Company, Shanghai, China), montmorillonite ((Al2,Mg3)Si4O10(OH)2•nH2O, 98%, Aladdin Reagent Company, Shanghai, China) and synthetic hematite, goethite and ferrihydrite." (Pages 5-6, Lines 107-110 in the revised manuscript)

-Line 128. How were the concentrations of minerals (5, 10, 25 mg/ml) decided? Did you measure the minimum inhibitory and minimum bactericidal concentrations for the J12 bacteria under the pH conditions of the experiment? Were the minerals only hydrated by the growth media? If so, what is the speciation of soluble metals with the components of the media solution?

R: The concentrations of minerals (5, 10, 25 mg/ml) were referred to the previous literature (McMahon et al., 2016). According to McMahon's results, negligible impacts of clays on bacteria growth when clays concentration is 5 mg/ml, while clays in suspensions exceeding 10 mg/ml in concentration inhibit the growth of bacteria. In this study, we did not measure the minimum inhibitory and minimum bactericidal concentrations for the J12 bacteria. The minerals were hydrated by both the growth media and the J12 bacteria. Our results showed that the speciation of soluble metals was soluble Al in the kaolinite and montmorillonite treatments (Figure S9) but both Fe(II) and Fe(III) in hematite, goethite and ferrihydrite treatments (Figure 4).

-Line 135. It is unclear why the particle size distribution is presented before and after incubation with bacteria. The increase in particle size after incubation (which would be better demonstrated in a graph than a table) probably results from agglomeration of mineral-bacteria clusters rather than a crystal growth. Is this important to the conclusions? If anything, a measurement of specific surface area of the minerals would be more important to the chemical interactions.

R: Good comment! In the revised manuscript, we replaced table S1 by Figure S4 to show the particle size distribution. We agree with the comment that the increase in particle size after incubation probably results from agglomeration of mineral-bacteria clusters, which was added in the Caption of Figure S4 in the revised manuscript. We are sorry for not measure the specific surface area of the minerals. However, according to the data provided by manufacturers, the specific surface area of kaolinite and mont- morillonite are ∼40 and 800 m2 g-1, respectively. The synthetic hematite, goethite and ferrihydrite were referred to the method from Schwertmann and Cornell (2007), and their specific surface area are approximately 30, 20, 200-300 m2 g-1, respectively. In the revised manuscript, we added the specific surface area of minerals and also listed as follows.

"According to the data provided by manufacturers, the specific surface area of kaolinite and montmorillonite are ∼40 and 800 m2 g-1, respectively. The synthetic hematite, goethite and ferrihydrite were referred to the method from Schwertmann and Cornell (2007), and their specific surface area are approximately 30, 20, 200-300 m2 g-1, respectively." was added in the revise manuscript (Page 7, Lines 160-164 in the revised

manuscript)

Figure S4. The particle size distribution (% in volume) of both the applied raw minerals and the changes after 24 h of cultivation. 1-11 represent the particle size of < 0.1, 0.1-0.5, 0.5-1, 1-2, 2-5, 5-10, 10-20, 20-50, 50-100, 100-500 $\mu$m, respectively. (a) kaolinite; (b) montmorillonite; (c) hematite; (d) goethite; (e) ferrihydtire; (f) kaolinite + bacteria; (g) montmorillonite + bacteria; (h) hematite + bacteria; (i) goethite + bacteria; (j) ferrihydtire + bacteria. The increase in particle size after incubation probably results from agglomeration of mineral-bacteria clusters.

-Line 209. For chemical analysis you have filtered the mineral-microbe suspension through 0.45 $\mu$m membrane, which may remove the bacteria, but allows clay size particles through. This is then analyzed by ICP and results reported as 'soluble' Al and Fe. However, the clay particles in this fraction will contribute to the elemental analysis.

R: Yes. Here 'soluble' Al and Fe should include the nano-size mineral particles. Suspension filtered through 0.45 $\mu$m membrane can be used to chemical analysis in mineral-microbial and soil systems (Ahmed and Holmström 2015; Li et al., 2018). And elements in the obtained solution were considered as dissolved.

Ahmed, E., and Holmström, S.J.M.: Microbe–mineral interactions: The impact of surface attachment on mineral weathering and element selectivity by microorganisms. Chem. Geol. 403, 13-23, 2015.

Li, Z. B., Lu, X., Teng, H. H., Chen, Y., Zhao, L., Ji, J., Chen, J., and Liu, L.: Specificity of low molecular weight organic acids on the release of elements from lizardite during fungal weathering. Geochim. Cosmochim. Acta Doi:10.1016/j.gca. 2018, in press, 2018.

-Line 225. This analysis presumes that OD600 only reflects absorbance by bacteria, but what is the absorbance of the mineral suspension alone?

R: We added the absorbance of the mineral suspension alone in the different concentrations. The results were added as Table S1 in the revised manuscript and shown as follows.

"Then, 50 $\mu$L of the cultures were transferred to fresh medium (10 mL) so that the effects of minerals were negligible (Table S1)." (Pages 6-7, Lines 130-131 in the revised manuscript)

Table S1. Optical density at 600 nm (OD600) of the mineral suspension (n = 3)

Mineral Absorbance at 600 nm 5 mg/mL 10 mg/mL 25 mg/mL Kaolinite 0.005 $\pm$ 0.001 0.019 $\pm$ 0.004 0.035 $\pm$ 0.003 Montmorillonite 0.010 $\pm$ 0.005 0.005 $\pm$ 0.001 0.017 $\pm$ 0.003 Hematite 0.004 $\pm$ 0.001 0.004 $\pm$ 0.001 0.004 $\pm$ 0.001 Goethite 0.008 $\pm$ 0.001 0.023 $\pm$ 0.002 0.037 $\pm$ 0.002 Ferrihydrite 0.002 $\pm$ 0.000 0.007 $\pm$ 0.003 0.015 $\pm$ 0.002

-Line 245. Mn(II), being redox active, is more likely to produce hydroxyl radical than to scavenge (Zarate-Reyes et al., 2017 Appl Clay Sci; Shi et al, 2016 Nature Sci Rev.)

R: We agree with the comment. Mn(II) can not only act as a scavenger for hydroxyl radical but also likely to produce hydroxyl radical (Lemire et al., 2013; Barnese et al., 2012). Because enzymes are vulnerable to the Fenton reaction and the produced hydroxyl radical, the presence of manganese may prevent protein damage by radicals (Anjem et al., 2009). In the revised manuscript, we revised this sentence and also listed as follows.

"The Mn(II) in the montmorillonite was reported to act as a scavenger for any hydroxyl radical production (Garrido-Ramírez et al., 2010), which may explain the promotion of microbial growth." (Page 12, Lines 244-246 in the original manuscript)

was changed to

"The Mn(II) in the montmorillonite was reported to act as a protective agent for enzymes (Anjem et al., 2009; Garrido-Ramírez et al., 2010), which may explain the promotion of microbial growth." (Page 12, Lines 253-255 in the revised manuscript) The added references are also listed as follows.

Anjem, A., Varghese, S., and Imlay, J. A.: Manganese import is a key element of the OxyR response to hydrogen peroxide in Escherichia coli. Mol. Microbiol. 72, 844–858, 2009.

Barnese, K., Gralla, E. B., Valentine, J. S. & Cabelli, D. E.: Biologically relevant mechanism for catalytic superoxide removal by simple manganese compounds. Proc. Natl. Acad. Sci. 109, 6892–6897, 2012.

Lemire, J. A.; Harrison, J. J.; Turner, R. J., Antimicrobial activity of metals: mechanisms, molecular targets and applications. Nat. Rev. Microbiol., 11 (6), 371-384, 2013.

-Line 267. Replace expect with except.

R: Thanks! In this sentence "expect" was changed to "except" (Page 13, Line 280 in the revised manuscript).

-Line 270. The production of acids by bacteria is plausible, but the surface area of the minerals is far greater (in general) than that of the bacteria, thus you might consider that each mineral surface attracts or repels H+ and OH- which is the major factor in buffering the fluid pH. You should test this by monitoring the pH (and Eh) of the suspension of minerals alone, compared to the suspension of bacteria alone.

R: In the revised manuscript, we added the evidence about the production of acids by providing the mapping of 3344 cm-1 (OH from -COOH) and 1230 cm-1 (C-O from -COOH) in Fig. 5. Based on the suggestion of Reviewer 1, we added the Eh of the suspension of minerals alone and the suspension of bacteria alone as Table S2. All experiments were performed in triplicate. The results from Eh showed that the Eh of minerals or bacteria alone was markedly decreased after 12 h cultivation (Table S2), suggesting an increase of proton production or cation solubility. The revised part was colored in red in the revised manuscript and also listed as follows.

"The pH decline suggests the production of organic acids by Pseudomonas brassicacearum J12." (Page 13, Lines 268-269 in the original manuscript)

was changed to

"The pH decline suggests the production of organic acids by Pseudomonas brassicacearum J12, which was supported by the presence of -OH (3344 cm-1) and C-O (1230 cm-1) from -COOH (Fig. 5). The redox potential (Eh) of minerals or bacteria alone was markedly decreased after 12 h cultivation (Table S2), suggesting an increase of proton production or cation solubility." (Page 13, Lines 281-286 in the revised manuscript)

Table S2. Redox potential (Eh) of the suspension of minerals alone, bacteria alone, and after incubation for 12 h (n = 3).

Treatment Eh (mV) Before incubation (mineral alone) After incubation (+ bacteria) Control 190.6 ± 20.18 110.3 ± 16.21 Kaolinite 135.1 ± 34.27 90.8 ± 10.84 Montmorillonite 142.5 ± 15.62 117.1 ± 25.17 Hematite 306.5 ± 43.74 189.5 ± 21.65 Goethite 199.4 ± 41.27 139.1 ± 11.17 Ferrihydrite 103.3 ± 5.88 79.8 ± 12.17

- Line 289. This requires H2O2, what is the source of that in the mineral-microbe suspension?

R: In this study, bacterium (i.e., Pseudomonas brassicacearum J12) is the source of H2O2.

-Line 295. Al3+ is not redox active, so why (or how) could it be correlated with production of hydroxyl radical? The toxicity of Al is from interactions with phospholipids, not production of radicals.

R: We agree with the comment. The toxicity of Al is from interactions with phospholipids, not production of radicals. Correlation between soluble Al and HO• (R = -0.35, t = -3.36, p = 0.004) was found, because the volume of sample used by test is little, under this condition no correlation between soluble Al and HO• is considered. In revised manuscript, we replaced "a weak" with "almost no" (Page 14, Line 311 in the revised manuscript).

-Line 305. This analysis is confusing to me. Did you add all of the minerals to this bacterial suspension and you are looking for which mineral dominates?

R: No. We did not add the minerals to this bacterial suspension. Here the sample was from the ferrihydrite treatment, i.e., cultivating 25 mg/mL ferrihydrite with Pseudomonas brassicacearum J12 for 12 h. During cultivation, iron oxides can be transformed into hematite, goethite, ferrihydrite, iron(II) oxalate, and iron(III) oxalate by the bacteria J12. Therefore, we used these samples to fit the component of iron oxides after cultivation.

-Line 320. This suggests that the bacteria reduce the mineral Fe, which in turn produces radicals that oxidize the bacteria. Why would bacteria not have a defense against such radicals?

R: Great comment! The production of ROS-scavenging enzymes, such as superoxide dismutase (SOD) and catalase (CAT), as well as other cellular antioxidants by bacteria may contribute to the protection of bacteria from such radicals. While excessive ROS in aerobic conditions causes the depletion of antioxidants and the inhibition of particular enzyme activities that are vital for cell growth (Lemire et al., 2013). Extracellular HO• oxidizes cardiolipin (CL), the important component of cell membrane, which would facilitate soluble $Fe^{2+}$ penetration into the cell that promotes intracellular HO• production (Wang et al., 2017). Thus, we deduce that HO• is continuously produced in the culture system and leads to oxidative damage of bacterial cells.

Lemire, J. A., Harrison, J. J., and Turner, R. J.: Antimicrobial activity of metals: mechanisms, molecular targets and applications. Nat. Rev. Microbiol., 11, 371-384, 2013.

Wang, X., Dong, H., Zeng, Q., Xia, Q., Zhang, L., and Zhou, Z.: Reduced iron-containing clay minerals as antibacterial agents, Environ. Sci. Technol., 51, 7639-7647, 2017.

- Line 341. A simple measurement of the OD600 on a mineral suspension should answer this definitively.

R: Thanks! We have added the absorbance of the mineral suspension alone as Table S1 in the revised manuscript. The revised part was colored in red and also listed as follows.

"As a result, the effect of mineral concentration may be minimal." (Page 16, Lines 340-341 in the original manuscript)

was changed to

"As a result, the effect of mineral concentration was minimal (Table S1)" (Page 14, Lines 366-367 in the revised manuscript)

Table S1. Optical density at 600 nm (OD600) of the mineral suspension (n = 3)

Mineral Absorbance at 600 nm 5 mg/mL 10 mg/mL 25 mg/mL Kaolinite $0.005 \pm 0.001$ $0.019 \pm 0.004$ $0.035 \pm 0.003$ Montmorillonite $0.010 \pm 0.005$ $0.005 \pm 0.001$ $0.017 \pm 0.003$ Hematite $0.004 \pm 0.001$ $0.004 \pm 0.001$ $0.004 \pm 0.001$ Goethite $0.008 \pm 0.001$ $0.023 \pm 0.002$ $0.037 \pm 0.002$ Ferrihydrite $0.002 \pm 0.000$ $0.007 \pm 0.003$ $0.015 \pm 0.002$

-Line 345. The inhibitory concentration of metals is pH dependent, so unless the work of Illmer and Schinner was at the same pH of your experiment (after 12 hrs) then this reported concentration may not be relevant. You need to determine the MIC and MBC concentrations at the pH of your mineral-microbe mixture (after incubation).

R: Thanks! In the revised manuscript, we deleted "A previous study showed that a 58 $\mu$M ($\sim$1.6 mg L-1) concentration of Al has toxicological effects on Pseudomonas sp. (Illmer and Schinner, 1999)." (Page 16, Lines 370 in the revised manuscript)

-Line 385. The phyllosilicates generally have a negative charge on their more extensive basal surfaces. Positive charges are limited to broken edges of the structure. This may not be true for ferrihydrite and goethite. But remember that hydroxyl radicals only exist for 1ns, so for them to interact with bacteria, there must be an attractions between the microbe and mineral surface where the radical is generated.

R: Great comment! In the revised manuscript, we revised this sentence and also listed as follows.

"we deduced that HO·Ać may mainly generate on the mineral surface, partly due to the positive charge of mineral surface (Tombácz and Szekeres, 2006) but the negative charge of microbes(Jucket et al., 1996)." (Page 18, Lines 384-387 in the original manuscript)

was changed to

"we deduced that HO·Ać may mainly generate on the mineral surface, in view of the fact that superoxide does not diffuse far from the site of formation (Tang et al., 2013)." (Page 18, Lines 417-419 in the revised manuscript)

-Line 403. It is unclear what you mean by 'stabilization role' here. What does the EPS do to stabilize the ferrihydrite?

R: Ferrihydrite is meta-stable and often aged to other crystalline minerals (e.g., hematite). The presence of EPS can enter the network structure of minerals and thus prevent the formation of crystalline minerals (Braunschweig et al., 2013; Li et al., 2016). In the revised manuscript, we added the corresponding explanation for this "stabilization role" and also listed as follows.

"The stabilization role of EPS was mainly identified as its combination into the network structure of minerals, which prevents the formation of crystalline minerals (Braunschweig et al., 2013)" (Page 19, Lines 441-443 in the revised manuscript).

Braunschweig J., Bosch J., and Meckenstock R. U.: Iron oxide nanoparticles in geomicrobiology: from biogeochemistry to bioremediation, New Biotechnol., 30, 793-802, 2013.

-Line 412. This is an important result of this study!

R: Thanks!

-Line 434. Do you suggest that the H2O2 required is generated by the bacteria or by dissolved O2 in solutions? Have you monitored the Eh of the solutions?

R: Yes. Our hypothesis in this study is that bacterium-initiated free-radical mechanism (i.e., Fenton-like reactions) by producing H2O2, which inhibits the growth of bacteria through. The paper published in Science by Diaz et al. (2013) had shown that taxonomically and ecologically diverse bacteria from terrestrial environments were a vast source of superoxide (O2 ïij∎) and H2O2 (Diaz et al., 2013).

Diaz, J. M., Hansel, C. M., Voelker, B. M., Mendes, C. M., Andeer, P. F., and Zhang, T.: Widespread production of extracellular superoxide by heterotrophic bacteria, Science, 340, 1223-1226, 2013.

In the revised manuscript, we added Eh of the suspension of minerals alone and the suspension of bacteria-mineral mixture (after incubation) as Table S2. All experiments were performed in triplicate.

Table S2. Redox potential (Eh) of the suspension of minerals alone, bacteria alone, and after incubation for 12 h (n = 3).

Treatment Eh (mV) Before incubation (mineral alone) After incubation (+ bacteria) Control 190.6 ± 20.18 110.3 ± 16.21 Kaolinite 135.1 ± 34.27 90.8 ± 10.84 Montmorillonite 142.5 ± 15.62 117.1 ± 25.17 Hematite 306.5 ± 43.74 189.5 ± 21.65 Goethite 199.4 ± 41.27 139.1 ± 11.17 Ferrihydrite 103.3 ± 5.88 79.8 ± 12.17

-Supplementary Figures Figure S6. How do you explain the drop in pH from 7.2 even in the control suspension? Figure S8. Why does the control have soluble Al even though there are no minerals in it?

R: Thanks! The drop in pH is mainly attributable to the fact that bacteria can secrete amounts of organic acids during incubation as shown in Fig. 5 in the revised manuscript. The concentration of soluble Al in Figure S8 is almost zero.

Please also note the supplement to this comment:
https://www.biogeosciences-discuss.net/bg-2018-479/bg-2018-479-AC1-
supplement.zip
* * *
[Figure]

[Figure]

**Fig. 1.** Optical density at 600 nm (OD600) of 8-h-old Pseudomonas brassicacearum J12 sub-cultures taken after 12 h growth with different minerals and with no minerals (control). K, kaolinite; M, montm

[Figure]

**Fig. 2.** Wide scan EPR spectra of both the kaolinite and montmorillonite.

[Figure]

**Fig. 3.** Generation of hydroxyl radical (HO•) after 12 h growth of Pseudomonas brassi-cacearum J12 with different minerals and with no minerals (control). K, kaolinite; M, montmo-rillonite; H, hematite; G, goeth

[Figure]

**Fig. 4.** Iron chemistry (a-c) and its correlation with hydroxyl radical (HO•) (d-f) as well as optical density at 600 nm (OD600) (g-i). (a) soluble Fe. (b) total Fe. (c) Fe(II). (d) soluble Fe vs HO•. (e) tota

[Figure]

**Fig. 5.** Correlative micro X-ray fluorescence ($\mu$-XRF) and synchrotron-based Fourier transform infrared (SR-FTIR) analysis of the thin section from the cultures of the 25 mg/mL ferrihydrite treatment after 12 h

[Figure]

**Fig. 6.** (a) Fe 2p X-ray photoelectron spectroscopy (XPS) spectra of ferrihydrite samples, F+bacteria and F-bacteria; (b-c) Fe 2p 3/2 spectra of F+bacteria and F-bacteria, respectively, during the cultivation

Substrate availability
(Indicated)

HO·

H$_2$O$_2$

Inhibition
(Direct)

Fe(III)

Pseudomonas
brassicacearum
J12

Intracellular inhibition

🔵 Ferrihydrite 🔵 Surface Fe(II) ═══ Crystalline Fe mineral
🔵 Soluble Fe$^{3+}$ 🔵 Soluble Fe$^{2+}$ 🟢 Orginic carbon 🟢 Organic acid

**Fig. 7.** Schematic of the bacterial inhibition by Fe(III)-containing minerals through a free-radical mechanism.

---

## Referee Comment (RC2) · Anonymous Referee #2 · 30 Jan 2019

I. General comments: This study aimed to study (i) the impact of Al- and Fe-containing minerals (montmorillonite, kaolinite, hematite, goethite and ferrihydrite) on bacterial growth using cultural approach on Pseudomonas brassicacearum J12 and (ii) the involvement of ROS, produced via fenton reactions, on Pseudomonas brassicacearum J12 growth. The subject is clearly interesting and is in accordance with researches published in Biogeosciences journal. Such researches on interactions between biotic and abiotic compartments are essential for our understanding of nutrients fluxes in soils and I encourage the publication of this manuscript in Biogeosciences journal. However, some points need to be clarified before publication.

[Figure]

II. Major comments:

- Major comment 1: Tilte: "Iron minerals inhibit the growth of bacteria via a free-radical mechanism: Implication for soil carbon storage": You cannot generalize your results to the domain of bacteria. I recognize that we will never be satisfied enough with the number of species studied, but I think that before expanding your results to the domain of bacteria, you should confirm them on other species from different phylum which show important genetic and phenotypic distances.

- Major comment 2: Integrate your statistical results in the description of the results and in your figures.

III. Specific comments:

Abstract

- "Together, these findings indicate that the reduced surface Fe(II) derived from Fe(III)-containing minerals inhibit bacteria via a free-radical mechanism, which may further contribute to soil carbon storage." : see Major comment 1. Free-radicals may lead to organic matter degradation-mineralization, you do not develop this idea in the manuscript.

Introduction

- This is a clear introduction which provide a good representation to the overall situation.

- "The bacterial inhibition property of a mineral is associated with the particular chemistry and with the mineral properties, 45 resulting in the various bacterial inhibition mechanisms of minerals" [l. 43-45]: can you please give more precisions on the various inhibition mechanisms?

- Please, name the Al- and the Fe-contaning minerals that you used in this study, it is hard to understand for non-chemists to which category belong kaolinite, montmorillonite, hematite, goethite, ferrihydrite.

Material and Methods

- I suggest to separate the paragraphe 2.1 into two parts: "2.1 Mineral preparation" [l.106], "2.2 Pseudomanas cultivation experiments" [l.121]

- Suppress "which is a major group of rhizobacteria that aggressively colonize plant roots, has been considered an important group for sustainable agriculture" [l.122-123]: the information is already given [l.90].

- Why didn't you chose to have a control [NB + Mineral]? The OD of this control can be subtracted from the OD measured in [NB + Mineral + Bacteria] and give you the OD of your bacteria without the disturbance induced by the mineral?

- Did you measure the kinetic of bacterial growth during the 12h? Are you sure that the bacteria is still in exponential phase of growth? Why did you chose 12h for the first incubation and 8h for the second one?

- pH measurement should be explain in "2.6 Chemical analysis"

- I do not understand the choice of an ANOVA, when did you used that test.

- Figure 1, 3, S5, S6, S7, S8 should integrate your statistical analysis.

- Explain/describe the "one-sample Kolmogorov-Smirnov Test"

- Which software did you use to find and represent the model that best fits with your data (Fig.4)?

Results

- In this part you should not interpret your results: [l.230-231], [l.244-247], [l.294-295], [l.268-269], [l.280-281], [l.313], [l.318-319], [l.327-328].

- Here, we are waiting for an exhaustive description of the results obtained during the study: give some values (mean $\pm$ SE) and precise when values are significantly (or not) different between the different conditions tested.

Paragraph [l.224-234]:

- "3.1. Bacterial inhibition by minerals" [l.223]: This title does not correspond in case of montmorillonite. I suggest something like: "Effect of mineral nature and their concentrations on P. brassicacearum J12 development".

- Suppress "The effects of the nature and content of tested minerals on the OD 600 of Pseudomonas brassicacearum J12 subcultures taken after 12 h growth are shown in Fig. 1.": it should be in the "Material and Methods" part.

- "Compared to Control (i.e., no minerals), the presence of montmorillonite significantly increased OD 600 .": give values.

- Suppress "On the other hand," [l. 227]

- [l.227-230]: "Presence of all other investigated minerals decreased OD 600 in the following order: ferrihydrite > goethite > hematite > kaolinite at 5 and 25 mg mL -1, and ferrihydrite > goethite > kaolinite > hematite at 10 mg mL -1": Please give some values.

- Suppress "Meanwhile" [l.232]

- "An increase in mineral concentration resulted in a significant decrease in OD 600, except for montmorillonite" [l. 232-233]: Give some values.

- Suppress "as the OD 600 seemed to be independent of its concentration" [l.233-234]: it is an interpretation. You can replace it by something like: "However, in presence of montmorillonite the OD600 is stable at 0.43 $\pm$ SE for all the mineral concentration studied"

- Fig.1: we do not see bottom bar of the SE

- Fig.1 text/description: you should mention the mineral concentrations.

Paragraph [l.235-247]:

- It represents a new idea: you should give it a title (e.g. "chemical structure of minerals")

- [l.235-247]: why didn't you describe the EPR profiles of ferrihydrite, goethite and hematite?

Paragraph [l.249-258]:

- "A 12 h cultivation of Pseudomonas brassicacearum J12 in the presence of different minerals revealed that generation of HO• 250 radicals in the cases of montmorillonite, kaolinite and hematite was almost similar to the control (Fig. 3)": "Almost"? You should precise if the difference are significant or not. To precise my comment, you should study the significance of the difference between the control and montmorillonite for the three concentrations, and kaolinite 25 mg.mL-1. Moreover, I think that the difference is significant between (i) montmorillonite 25mg.mL-1 and kaolinite 25mg.mL-1 and (ii) montmorillonite 25mg.mL-1 and hematite 25mg.mL-1.

- [l255]: replace "rapidly" by significantly: there is no notion of time.

- Fig.3 text/description: you should mention the mineral concentrations

Paragraph [l.259-295]:

- Globally, I encourage the authors to reorganize this part of the manuscript. You should describe all your results (Fig. 4) not only those which are consistent with your interpretation. Just for example: I observe a significant increase of soluble Fe in the treatment containing goethite with the increase of goethite concentration but this results is missing from the test.

- Given the importance of the pH in the results description, I think that this result may be integrated in Fig.4.

- [l. 261-263]: "Much more soluble Fe was released from Fe(III)-containing minerals than from montmorillonite, kaolinite, and control (Fig. 4a)": Please give some values.

- "The solubility of Fe is closely related to pH value.": Are you sure about that? The pH of goethite solution is equivalent to the pH of kaolinite, montmorillonite, hematite and

goethite but the solubility of Fe in solution containing hematite and goethite seems to be more important. You should draw the graph showing the correlation between pH and the soluble Fe.

- Fig.4.b: You have a surprising result: the significant decrease of Total Fe in solutions containing 25 mg.mL-1 ferrihydrite between 2h and 12h. How do you explain that?

- "For all of the examined minerals, the trends of total Fe and Fe(II) were similar in the following order: ferrihydrite » goethite > hematite > montmorillonite ≈ kaolinite ≈ control (Fig. 4b-4c)": Please give some values. What do you mean with » and ≈? Is there a relation with a statistical analysis?

- [l.274-245] "Furthermore, a positive correlation exists between OD600 and soluble Fe content 275 (R = 0.92, t = -3.49, p = 0.003) and Fe(II) (R = 0.98, t = -4.28, p =0.001) (Fig. 4d and 4f, Table S2).": I think that you wanted to say "a positive correlation between Hydroxyl radical content and soluble Fe content".

- The interpretation of "R" and "t" should appear in the Material and Methods.

- [l.277]: "R=-0.75" and "t= 2.27" do not correspond to the values in the Fig.4.

- Fig.S7 should appear in Fig.4

- Fig.S7: Can you explain why the inhibition of Pseudomonas is more important in Fe(III) 100mg.L-1 than in Fe(III) 50mg.L-1 while hydroxyl radical production still the same between those two concentrations? Is that not the sign of the existence of another process involved in the Pseudomonas growth inhibition? I find this result very important, it should be considered in your discussion.

Paragraph [l.296-328]:

For non-chemist, this part is difficult to understand. Maybe the next comments will allow you to make it more accessible for biologists.

- Fig.5.a: What do the colors mean?

- Why did you select those regions of the spectra for XANES?

- Fig.5.c: What do the colors mean?

- Conserve the same colors between Fig.5.b Spot A and B.

- [l.309]: Spot A or Spot B?

- [l.307-309]: why don't you speak about FeC2O4 (25.9%) in spot B?

- Why is there goethite and hematite in sample which only contain ferrihydrite?

- You should give different title to the paragraph [l.298-309], [l.310-320]

- Paragraph [l.321-328] should be describe in paragraph [l.298-309]: it is the same figure and consequently the same idea.

- [l.317-320] "Interestingly, the area of the peak at 709.5 eV was bigger in the F + bacteria treatment than that in F - bacteria treatment (Fig. 6b-6c), suggesting that Fe(II) was generated on the surface of ferrihydrite during the cultivation with bacteria. Based on the reaction 1, HO2 • should be the oxidant products.": Is that reproducible between samples? Is that spectrum the mean representation of several spectra?

- Fig.6 b and c: what do the colors mean?

- Fig.6 a: Correct "Inyensity" by "Intensity"

- [l.325]: "good", can you precise this term please?

New paragraph:

- Given the importance of Al in your discussion (half of the discussion), the Fig.S8 should appear in the main manuscript (not in SupMat) with the results presented in Fig.4.

- In Fig.1, Fig.3, Fig.4, Fig.S6, Fig.S8: you should distinguish Al- from Fe- containing minerals.

Discussion

Another time: I disapprove the use of the term "bacteria" which may refer to the domain of bacteria (see main comment 1).

4.1. Effect of Al(III)-containing minerals on the inhibition of bacterial growth

- [l.336-343]: "It should be noted that the presence of minerals may potentially interfere with the measurement of cell numbers in Fig. 1. In this study, we subsampled the experimental cultures and diluted them in fresh medium so that both clay particles and bacteria were 200× less concentrated (Fig. S3), following the protocol of McMahon et al. (2016). As a result, the effect of mineral concentration may be minimal. In addition, plating the bacteria by evaluating populations by counting colonies may act as a complementary method for OD600 and needs to be investigated in the future.": I am not waiting for a response to that comment: In your case, I would have chosen the association of a cell labeling with DAPI and a count of labeled cells with flow cytometry (or fluorescence microscopy).

- [l.355-357]: "Furthermore, the formation of some Al intermediates by the decreasing pH, such as Al13O4(OH)24 7+, is also suggested to be more toxic for bacterial growth (Amonette et al., 2003; Liu et al., 2016).": what pH are you referring to? Is that in accordance with the pH measured in your study?

- The information given at [l.357-359] should appear after [l.345-349]. Then, you can discuss (i) on the results that you expected to observe and (ii) on the interpretation of the results that you obtained.

4.2. Inhibition of bacteria by Fe(III)-containing minerals via a free-radical mechanism

- If we take the two equations cited in your introduction: (1) ≡ Fe(III)-OH + H2O2 ≡ Fe(II) + H2O + HO2 (2) ≡ Fe(II) + H2O2 ≡ Fe(III)-OH + HO• Where does H2O2 come from? Pseudomonas? If it come from the bacteria, the reduction of it development should induce a decrease of HO• production in LB medium containing Fe

minerals (if H2O2 is the limiting compound in the reaction, and it should be the case here), am I wrong?

- Correct the sentence [l.383-387]: "In line with other studies (Kwan and Voelker, 2003; Wang et al., 2017b), we deduced that HO• may mainly generate on the mineral surface, partly due to the positive charge of mineral surface (Tombácz and Szekeres, 2006) but the negative charge of microbes (Jucket et al., 1996)."

- [l.401-404]: "High percentage of the less stable ferrihydrite (Table S3) may be attributable to the stabilization role of produced EPS (Fig. 5c) by bacteria to minerals, which had been shown during the cultivation of fungi with minerals (Li et al., 2016).Âż please, divide this sentence into two sentences in order to distinguish your contribution from the contribution of Li et al. (2016). Can you precise your idea on the role of EPS on stabilization process please?

- [l.414]: suppress "cellular". I do not understand the difference between cellular and free reductant? Free reductant such as FADH2 are intracellular, no? I think that you want to separate (i) cellular from (ii) non-cellular reactions, am I wrong?

- [l.413-418]: Are those reactions linked to HO• production?

- [l.424]: Given that results in Fig.1 and Fig.S7 are produce by different experimental device, are you sur that you can give this interpretation to your results?

- [l.427]: "simultaneously"?

4.3. Inhibition of bacterial growth by a free-radical mechanism and its implications for soil carbon storage

- Fig.7: Can you please explain the figure in the caption?

- Can you please go further in the processes through which soluble Fe3+ and Fe2+ will have an inhibition effect on Pseudomonas?

- Can you please go further in processes through which HO• will have a "direct"

inhibition power on Pseudomonas (modification of cell membrane physico-chemical properties?)

- [l.436-437]: "Oxidative damage of extracellular HO• may lead to bacterial inactivation, and protection of carbon from microbial degradation." Please go further in this interpretation: HO• have a role on Pseudomonas growth (it is your study), but HO• can have other impacts in soils. What are they? How can HO• and Fe act (i) on the soil C storage and (ii) on the soil C degradation-mineralization?

- [l.439-442]: "In addition, the generation of free radicals may also have indirect effects on bacterial growth via substrate availability (Table S4). Substrate availability is improved in the presence of radicals, owing to the following two facts: 1) the depolymerization role of radicals on the complex substrates; 2) the inhibition role of radicals on bacteria indirectly increasing the amounts of available substrates.": Do you think that we can see an inhibition of bacterial growth through the reduction of nutrient availability induced by free radicals in a medium where nutrients are in excess?

- What about the role of minerals on the "stabilization-adsorption" of organic compounds of the NB medium?

- [l.445-448]: Fe is one of the numerous processes regulating carbon cycle in soils. I suggest something like: "In this study, we suggest that soil carbon cycle is partly regulated by Fe minerals (i) by the formation of organo-mineral complexes and (ii) by the bacterial development inhibition (specify the processes)."

- l.451 replace "but" by "and"

Conclusions

- [l. 467-458]: "effects on bacterial growth and the presence of minerals may potentially interfere with the measurement of cell numbers": I do not think that it is necessary to speak about that here.

---

## Author Response (AR2)

**Reply to the Comments**

**Anonymous Referee #1**

**The authors appreciate the report of Referee #1 and respond as follows.**

General Comments: This study is a nice contribution to the analysis of mineral-microbe interactions in soils, and it presents some new evidence for the role of iron in diminishing certain bacterial populations. Of particular interest to me was the X-ray photoelectron spectroscopy data demonstrating the presence of Fe(II) on the ferrihydrite surface, suggesting that the microbe (J12) reduces mineral bound Fe(III). This is important, because Fe(II) is then available for Fenton reactions that can produce radicals that damage cell membranes allowing soluble metals to enter the bacterial cell. The work suggests that iron bearing minerals in soils contribute to the preservation of organic C by limiting the productivity of bacteria that degrade carbon.

**Specific Comments:**

-Line 108. Your description of the clay minerals used is not adequate for determining the chemical composition and stability of the phyllosilicates. In particular montmorillonite has many chemical components in each crystallographic site that contribute to the mineral surface characteristics (charge distribution) and interlayer cations. Therefore, one must give the chemical formula for the mineral used. One montmorillonite might increase bacterial growth (if it provides nutrients) while another might decrease bacterial growth (if it provides toxins).

**Response (R)**: Agree! In the revised manuscript, the chemical formulas for the clay mineral used were added and also listed as follows.

"Five minerals were selected in this study, including kaolinite (98%, Aladdin Reagent Company, Shanghai, China), montmorillonite (98%, Aladdin Reagent Company, Shanghai, China) and synthetic hematite, goethite and ferrihydrite." (**Pages 5-6, Lines 107-109 in the original manuscript**)

**was changed to**

"Five minerals were selected in this study, including kaolinite  $(Al_2O_3.2SiO_2.2H_2O, 98\%, Aladdin Reagent Company, Shanghai, China), montmorillonite ((Al_2,Mg_3)Si_4O_{10}(OH)_2.nH_2O, 98\%, Aladdin Reagent Company, Shanghai, China) and synthetic hematite, goethite and ferrihydrite." (Page 6, Lines 126-129 in the revised manuscript)$

-Line 128. How were the concentrations of minerals (5, 10, 25 mg/ml) decided? Did you measure the minimum inhibitory and minimum bactericidal concentrations for the J12 bacteria under the pH conditions of the experiment? Were the minerals only

hydrated by the growth media? If so, what is the speciation of soluble metals with the components of the media solution?

**R:** The concentrations of minerals (5, 10, 25 mg/ml) were referred to the previous literature (McMahon et al., 2016). According to McMahon's results, negligible impacts of clays on bacteria growth when clays concentration is 5 mg/ml, while clays in suspensions exceeding 10 mg/ml in concentration inhibit the growth of bacteria. The minerals were only hydrated by the growth media. Our results showed that the speciation of soluble metals was soluble Al in the kaolinite and montmorillonite treatments (Figure 6) but both Fe(II) and Fe(III) in hematite, goethite and ferrihydrite treatments (Figure 5).

-Line 135. It is unclear why the particle size distribution is presented before and after incubation with bacteria. The increase in particle size after incubation (which would be better demonstrated in a graph than a table) probably results from agglomeration of mineral-bacteria clusters rather than a crystal growth. Is this important to the conclusions? If anything, a measurement of specific surface area of the minerals would be more important to the chemical interactions.

**R:** Good comment! In the revised manuscript, we replaced table by graph (as Figure S4) to show the particle size distribution. Mineral binding is a major mechanism for carbon (C) stabilization. We agree with the comment that the increase in particle size after incubation probably results from agglomeration of mineral-bacteria clusters, which was added in the Discussion section in the revised manuscript. We are sorry for not measure the specific surface area of the minerals. However, according to the data provided by manufacturers, the specific surface area of kaolinite and montmorillonite are ~40 and 800 m2 g-1, respectively. The synthetic hematite, goethite and ferrihydrite were referred to the method from Schwertmann and Cornell (2007), and their specific surface area are approximately 30, 20, 200-300 m2 g-1, respectively. In the revised manuscript, we added the specific surface area of minerals and also listed as follows.

"According to the data provided by manufacturers, the specific surface area of kaolinite and montmorillonite are ~40 and 800 m2 g-1, respectively. The synthetic hematite, goethite and ferrihydrite were referred to the method from Schwertmann and Cornell (2007), and their specific surface area are approximately 30, 20, 200-300 m2 g-1, respectively." was added in the revise manuscript (**Page 7, Lines 165-169 in the revised manuscript**)

**Figure S4.** The particle size distribution (% in volume) of both the applied raw minerals and the changes after 24 h of cultivation. 1-11 represent the particle size of  $< 0.1, 0.1-0.5, 0.5-1, 1-2, 2-5, 5-10, 10-20, 20-50, 50-100, 100-500 \,\mu\text{m}$ , respectively. (a) kaolinite; (b) montmorillonite; (c) hematite; (d) goethite; (e) ferrihydtire; (f) kaolinite + bacteria; (g) montmorillonite + bacteria; (h) hematite + bacteria; (i) goethite + bacteria; (j) ferrihydtire + bacteria.

-Line 209. For chemical analysis you have filtered the mineral-microbe suspension through 0.45  $\mu$ m membrane, which may remove the bacteria, but allows clay size particles through. This is then analyzed by ICP and results reported as 'soluble' Al and Fe. However, the clay particles in this fraction will contribute to the elemental analysis.

**R:** Yes. Here 'soluble' Al and Fe should include the nano-size mineral particles. Suspension filtered through 0.45  $\mu$ m membrane can be used to chemical analysis in mineral-microbial and soil systems (Ahmed and Holmström 2015; Li et al., 2018). And elements in the obtained solution were considered as dissolved.

- Ahmed, E. and Holmström, S.J.M.: Microbe–mineral interactions: The impact of surface attachment on mineral weathering and element selectivity by microorganisms. Chem. Geol. 403, 13-23, 2015.
- Li, Z.-b., Lu, X., Teng, H.H., Chen, Y., Zhao, L., Ji, J., Chen, J. and Liu, L.: Specificity of low molecular weight organic acids on the release of elements from lizardite during fungal weathering. Geochim. Cosmochim. Ac. doi:10.1016/j.gca. 2018.

-Line 225. This analysis presumes that  $OD_{600}$  only reflects absorbance by bacteria, but what is the absorbance of the mineral suspension alone?

**R:** We had detected the absorbance of the mineral suspension alone and diluted 200 times mineral suspension. The results were added as Table S1 in the revised manuscript and shown as follows.

|                 | Absorbance at 600 nm |                   |                 |                   |                   |                   |  |
|-----------------|----------------------|-------------------|-----------------|-------------------|-------------------|-------------------|--|
| Mineral         | Mineral suspension   |                   |                 |                   | Diluted 200 times |                   |  |
|                 | 5 mg/mL              | 10 mg/mL          | 25 mg/mL        | 5 mg/mL           | 10 mg/mL          | 25 mg/mL          |  |
| Kaolinite       | $1.625\pm0.059$      | $1.798 \pm 0.047$ | $2.046\pm0.023$ | $0.005\pm0.001$   | $0.019\pm0.004$   | $0.035\pm0.003$   |  |
| Montmorillonite | $1.196\pm0.047$      | $1.319\pm0.017$   | $1.802\pm0.025$ | $0.010\pm0.005$   | $0.005\pm0.001$   | $0.017\pm0.003$   |  |
| Hematite        | $2.856\pm0.085$      | $2.977\pm0.041$   | $2.942\pm0.062$ | $0.004\pm0.001$   | $0.004 \pm 0.001$ | $0.004 \pm 0.001$ |  |
| Goethite        | $2.228\pm0.071$      | $2.485\pm0.052$   | $2.703\pm0.057$ | $0.008 \pm 0.001$ | $0.023\pm0.002$   | $0.037\pm0.002$   |  |
| Ferrihydrite    | $1.477\pm0.149$      | $1.999 \pm 0.179$ | $2.814\pm0.034$ | $0.002\pm0.000$   | $0.007 \pm 0.003$ | $0.015\pm0.002$   |  |

**Table S1.**  $OD_{600}$  of the mineral suspension (n = 3).

-Line 245. Mn(II), being redox active, is more likely to produce hydroxyl radical than to scavenge (Zarate-Reyes et al., 2017 Appl Clay Sci; Shi et al, 2016 Nature Sci Rev.)

**R:** We agree with the comment. Mn(II) can act as a scavenger for hydroxyl radical and is also likely to produce hydroxyl radical. According to Lemire et al (2013) and Barnese et al (2012), although Mn(II) also catalyses Fenton chemistry in vitro, accumulating evidence suggests that this is not the case in vivo and, in fact, that Mn functions in protecting cells from ROS. Furthermore, bacteria increases its uptake of Mn(II) that protecting mononuclear metalloenzymes from destruction by other toxic metals (Anjem et al., 2009). In the revised manuscript, we deleted "The Mn(II) in the montmorillonite was reported to act as a scavenger for any hydroxyl radical production (Garrido-Ramírez et al., 2010), which may explain the promotion of microbial growth." (Page 13, Line 355 in the revised manuscript)

- Anjem, A., Varghese, S., and Imlay, J. A.: Manganese import is a key element of the OxyR response to hydrogen peroxide in *Escherichia coli*. Mol. Microbiol. 72, 844–858, 2009.
- Barnese, K., Gralla, E. B., Valentine, J. S. & Cabelli, D. E.: Biologically relevant mechanism for catalytic superoxide removal by simple manganese compounds. Proc. Natl. Acad. Sci. 109, 6892–6897, 2012.
- Lemire, J. A.; Harrison, J. J.; Turner, R. J., Antimicrobial activity of metals: mechanisms, molecular targets and applications. Nat. Rev. Microbiol., 11 (6), 371-84, 2013.
- -Line 267. Replace expect with except.
- **R:** Thanks! In this sentence "expect" was changed to "except" (**Page 14, Line 405 in the revised manuscript**).

-Line 270. The production of acids by bacteria is plausible, but the surface area of the minerals is far greater (in general) than that of the bacteria, thus you might consider

that each mineral surface attracts or repels  $H^+$  and  $OH^-$  which is the major factor in buffering the fluid pH. You should test this by monitoring the pH (and Eh) of the suspension of minerals alone, compared to the suspension of bacteria alone.

**R:** Great comment! In the revised manuscript, we detected the Eh of the suspension of minerals alone (25 mg/ml) and of bacteria-mineral mixture and added in the SI as Table S5. All experiments were performed in triplicate. Results showed that Eh of bacteria-mineral mixture after incubation was generally lower than the suspension of minerals alone, suggesting that the redox potential was decreased by the interaction between mineral and bacteria. Redox potential is a major factor that influencing cation availability (Cheng et al., 2010). Compared to hematite and goethite, ferrihydrite is more reactive and Fe3+ can be readily converted to Fe2+ under low redox potential, thereby enhancing the iron solubility. Furthermore, acidic cations such as H+ also contributed to the dissolution of cations (Cheng et al., 2010). Thus, in the revised manuscript we added the Eh results and also listed as follows.

"The solubility of Fe is closely related to pH value. Therefore, the solution pH was determined after 12 h growth of *Pseudomonas brassicacearum* J12 with different minerals and with no minerals (control) (Fig. S6)." (Page 13, Lines 264-266 in the original manuscript)

**was changed to**

"The solubility of Fe was closely related to redox potential and pH value (Fig. S8). Results showed that Eh of bacteria-mineral mixture after incubation was generally lower than the suspension of minerals alone (Table S5), suggesting that the redox potential was decreased by the interaction between mineral and J12. Furthermore, the solution pH was determined after 12 h growth of J12 with different minerals and with no minerals (control) (Fig. 4)." (Page 14, Lines 399-405 in the revised manuscript)

Cheng, L., Zhu, J., Chen, G., Zheng, X., Oh, N. H., Rufty, T. W., Richter, D., Hu, S., Atmospheric CO2 enrichment facilitates cation release from soil, Ecol. Lett., 13 (3), 284-291, 2010.

|                 | Eh (1             | mV)               |  |
|-----------------|-------------------|-------------------|--|
| Treatment       | Before incubation | After incubation  |  |
|                 | (alone)           | (+ bacteria)      |  |
| Control         | $190.6 \pm 20.18$ | $110.3 \pm 16.21$ |  |
| Kaolinite       | $135.1 \pm 34.27$ | $90.8 \pm 10.84$  |  |
| Montmorillonite | $142.5 \pm 15.62$ | $117.1 \pm 25.17$ |  |
| Hematite        | $306.5 \pm 43.74$ | $189.5 \pm 21.65$ |  |
| Goethite        | $139.1 \pm 11.17$ | $199.4 \pm 41.27$ |  |
| Ferrihydrite    | $103.3 \pm 5.88$  | $79.8 \pm 12.17$  |  |

Table S5. Eh of the suspension of minerals alone (25 mg/ml) and of bacteria-mineral mixture

- Line 289. This requires  $H_2O_2$ , what is the source of that in the mineral-microbe suspension?

**R:** In this study, bacterium (i.e., *Pseudomonas brassicacearum* J12) is the source of  $H_2O_2$ .

-Line 295.  $Al^{3+}$  is not redox active, so why (or how) could it be correlated with production of hydroxyl radical? The toxicity of Al is from interactions with phospholipids, not production of radicals.

**R:** We agree with the comment. The toxicity of Al is from interactions with phospholipids, not production of radicals. Correlation between soluble Al and HO• (R = -0.35, t = -3.36, p = 0.004) was found, because the volume of sample used by test is little, under this condition no correlation between soluble Al and HO• is considered. In revised manuscript, we replaced "a weak" with "almost no" (**Page 15, Line 481 in the revised manuscript**).

-Line 305. This analysis is confusing to me. Did you add all of the minerals to this bacterial suspension and you are looking for which mineral dominates?

**R**: No. We did not add the minerals to this bacterial suspension. Here the sample was from the ferrihydrite treatment, i.e., cultivating 25 mg/mL ferrihydrite with *Pseudomonas brassicacearum* J12 for 12 h. In this treatment, iron oxides may be transformed into hematite, goethite, ferrihydrite, iron(II) oxalate, and iron(III) oxalate after cultivated with *Pseudomonas brassicacearum* J12. Therefore, we used these samples to fit the component of iron oxides after cultivation.

-Line 320. This suggests that the bacteria reduce the mineral Fe, which in turn produces radicals that oxidize the bacteria. Why would bacteria not have a defense against such radicals?

**R**: Great comment! The production of ROS-scavenging enzymes, such as superoxide

dismutase (SOD) and catalase (CAT), as well as other cellular antioxidants by bacteria may contribute to the protection of bacteria from such radicals. While excessive ROS in aerobic conditions causes the depletion of antioxidants and the inhibition of particular enzyme activities that are vital for cell growth (Lemire et al., 2013). Extracellular HO• oxidizes cardiolipin (CL), the important component of cell membrane, which would facilitate soluble  $Fe^{2+}$  penetration into the cell that promotes intracellular HO• production (Wang et al., 2017). Furthermore, our unpublished data showed that H2O2 reacts with Fe(III) minerals (i.e. hematite, goethite, and ferrihydrite) to form HO• as well. Thus, we deduce that HO• is continuously produced in the culture system and leads to oxidative damage of bacterial cells.

- Lemire, J. A., Harrison, J. J., and Turner, R. J.: Antimicrobial activity of metals: mechanisms, molecular targets and applications. Nat. Rev. Microbiol., 11, 371-384, 2013.
- Wang, X., Dong, H., Zeng, Q., Xia, Q., Zhang, L., and Zhou, Z.: Reduced iron-containing clay minerals as antibacterial agents, Environ. Sci. Technol., 51, 7639-7647, 2017.

- Line 341. A simple measurement of the  $OD_{600}$  on a mineral suspension should answer this definitively.

**R**: Thanks! We have detected the absorbance of the mineral suspension alone and diluted 200 times mineral suspension. The results were added as Table S1 in the revised manuscript and shown as follows.

|                 | Absorbance at 600 nm |                   |                   |                   |                   |                 |  |
|-----------------|----------------------|-------------------|-------------------|-------------------|-------------------|-----------------|--|
| Mineral         | Mineral suspension   |                   |                   |                   | Diluted 200 times |                 |  |
|                 | 5 mg/mL              | 10 mg/mL          | 25 mg/mL          | 5 mg/mL           | 10 mg/mL          | 25 mg/mL        |  |
| Kaolinite       | $1.625\pm0.059$      | $1.798 \pm 0.047$ | $2.046\pm0.023$   | $0.005\pm0.001$   | $0.019\pm0.004$   | $0.035\pm0.003$ |  |
| Montmorillonite | $1.196\pm0.047$      | $1.319\pm0.017$   | $1.802\pm0.025$   | $0.010\pm0.005$   | $0.005\pm0.001$   | $0.017\pm0.003$ |  |
| Hematite        | $2.856\pm0.085$      | $2.977\pm0.041$   | $2.942\pm0.062$   | $0.004\pm0.001$   | $0.004\pm0.001$   | $0.004\pm0.001$ |  |
| Goethite        | $2.228\pm0.071$      | $2.485\pm0.052$   | $2.703\pm0.057$   | $0.008 \pm 0.001$ | $0.023\pm0.002$   | $0.037\pm0.002$ |  |
| Ferrihydrite    | $1.477\pm0.149$      | $1.999 \pm 0.179$ | $2.814 \pm 0.034$ | $0.002\pm0.000$   | $0.007\pm0.003$   | $0.015\pm0.002$ |  |

**Table S1.**  $OD_{600}$  of the mineral suspension (n = 3).

-Line 345. The inhibitory concentration of metals is pH dependent, so unless the work of Illmer and Schinner was at the same pH of your experiment (after 12 hrs) then this reported concentration may not be relevant. You need to determine the MIC and MBC concentrations at the pH of your mineral-microbe mixture (after incubation).

 $\underline{\mathbf{R}}$ : Thanks! We agree with the comment that the inhibitory concentration of metals is pH dependent. Thus, in the revised manuscript, we deleted "A previous study

showed that a 58  $\mu$ M (~1.6 mg L-1) concentration of Al has toxicological effects on *Pseudomonas sp.* (Illmer and Schinner, 1999)." (**Page 17, Line 589 in the revised manuscript**)

-Line 385. The phyllosilicates generally have a negative charge on their more extensive basal surfaces. Positive charges are limited to broken edges of the structure. This may not be true for ferrihydrite and goethite. But remember that hydroxyl radicals only exist for 1ns, so for them to interact with bacteria, there must be an attractions between the microbe and mineral surface where the radical is generated.

**R**: Great comment! Mineral surface can provide transition metals and adsorb ROS-inducing compounds, where is favorable for the formation of HO\*. A striking capability that has been reported for many particles, especially nanoparticles, is their ability to physically interact with the cell surfaces of some bacteria (Stoimenov et al., 2002; Luef et al., 2012). According to Luef's study (2012), a three-dimensional reconstruction of a groundwater bacterium decorated with nanoparticles shows that nano-aggregates are located outside the cell wall. The toxic mode of action of nanoparticles has been associated with ROS generation and membrane disruption. Thus, we deduced that HO\* formed in the mineral surface reacts with bacteria immediately and leads to the inactivation of bacteria.

- Stoimenov, P. K., Klinger, R. L., Marchin, G. L., and Klabunde, K. J.: Metal oxide nanoparticles as bactericidal agents, Langmuir, 18, 6679-6686, 2002.
- Luef, B. et al.: Iron-reducing bacteria accumulate ferric oxyhydroxide nanoparticle aggregates that may support planktonic growth, ISME J., 7, 338-350, 2012.

-Line 403. It is unclear what you mean by 'stabilization role' here. What does the EPS do to stabilize the ferrihydrite?

**R**: Ferrihydrite is meta-stable and often aged to other crystalline minerals (e.g., hematite). The presence of EPS can enter the network structure of minerals and thus prevent the formation of crystalline minerals (Braunschweig et al., 2013; Li et al., 2016). In the revised manuscript, we added the corresponding explanation for this "stabilization role" and also listed as follows.

"The stabilization role of EPS was mainly identified as its combination into the network structure of minerals, which prevents the formation of crystalline minerals (Braunschweig et al., 2013)" (Page 20, Lines 698-701 in the revised manuscript).

Braunschweig J., Bosch J., and Meckenstock R. U.: Iron oxide nanoparticles in geomicrobiology: from biogeochemistry to bioremediation, New Biotechnol., 30, 793-802, 2013.

-Line 412. This is an important result of this study!

**R**: Thanks!**

-Line 434. Do you suggest that the  $H_2O_2$  required is generated by the bacteria or by dissolved  $O_2$  in solutions? Have you monitored the Eh of the solutions?

**R**: Yes. Our hypothesis in this study is that bacterium-initiated free-radical mechanism (i.e., Fenton-like reactions) by producing  $H_2O_2$ , which inhibits the growth of bacteria through. The paper published in Science by Diaz et al. (2013) had shown that taxonomically and ecologically diverse bacteria from terrestrial environments were a vast source of superoxide ( $O_2^{\bullet-}$ ) and  $H_2O_2$  (Diaz et al., 2013).

In the revised manuscript, we detected Eh of the suspension of minerals alone and the suspension of bacteria-mineral mixture (after incubation). All experiments were performed in triplicate. Results showed that Eh of bacteria-mineral mixture after incubation was generally lower than the suspension of minerals alone. Data are listed as follows.

|                 | Eh (t             | mV)               |  |  |
|-----------------|-------------------|-------------------|--|--|
| Treatment       | Before incubation | After incubation  |  |  |
|                 | (alone)           | (+ bacteria)      |  |  |
| Control         | $190.6 \pm 20.18$ | $110.3 \pm 16.21$ |  |  |
| Kaolinite       | $135.1 \pm 34.27$ | $90.8 \pm 10.84$  |  |  |
| Montmorillonite | $142.5 \pm 15.62$ | $117.1 \pm 25.17$ |  |  |
| Hematite        | $306.5 \pm 43.74$ | $189.5 \pm 21.65$ |  |  |
| Goethite        | $199.4 \pm 41.27$ | $139.1 \pm 11.17$ |  |  |
| Ferrihydrite    | $103.3\pm5.88$    | $79.8 \pm 12.17$  |  |  |

Table S5.

-Supplementary Figures Figure S6. How do you explain the drop in pH from 7.2 even in the control suspension? Figure S8. Why does the control have soluble A1 even though there are no minerals in it?

**R**: Thanks! The drop in pH is mainly attributable to the fact that bacteria can secrete amounts of organic acids during incubation. The concentration of soluble Al in Figure S8 (Figure 6 in the revised manuscript) is almost zero.

Diaz, J. M., Hansel, C. M., Voelker, B. M., Mendes, C. M., Andeer, P. F., and Zhang, T.: Widespread production of extracellular superoxide by heterotrophic bacteria, Science, 340, 1223-1226, 2013.

**Anonymous Referee #2**

**The authors appreciate the report of Referee #2 and respond as follows.**

I. General comments: This study aimed to study (i) the impact of Al- and Fe-containing minerals (montmorillonite, kaolinite, hematite, goethite and ferrihydrite) on bacterial growth using cultural approach on Pseudomonas brassicacearum J12 and (ii) the involvement of ROS, produced via fenton reactions, on Pseudomonas brassicacearumJ12 growth. The subject is clearly interesting and is in accordance with researches published in Biogeosciences journal. Such researches on interactions between biotic and abiotic compartments are essential for our understanding of nutrients fluxes in soils and I encourage the publication of this manuscript in Biogeosciences journal. However, some points need to be clarified before publication.

**II. Major comments:**

- Major comment 1: Title: "Iron minerals inhibit the growth of bacteria via a free-radical mechanism: Implication for soil carbon storage": You cannot generalize your results to the domain of bacteria. I recognize that we will never be satisfied enough with the number of species studied, but I think that before expanding your results to the domain of bacteria, you should confirm them on other species from different phylum which show important genetic and phenotypic distances.

**Response** (**R**): Thanks! In the revised manuscript, we revised "bacteria" to "*Pseudomonas brassicacearum* J12" in the Title. Diaz et al. (2013) showed that other species, e.g., *Pseudomonas putida* GB-1, could produce approximately 1 and 10 amol  $O_2^{\bullet^-}$  cell-1 h-1 during mid-exponential growth or stationary phase, respectively. Except for *Pseudomonas*, taxonomically and ecologically diverse heterotrophic bacteria from both aquatic and terrestrial environments were a vast source of superoxide ( $O_2^{\bullet^-}$ ) and  $H_2O_2$  (Diaz et al., 2013). Based on the suggestion of Referee #2 and the results from Diaz et al. (2013), we think that "iron minerals inhibit the growth of heterotrophic bacteria via a free-radical mechanism" should be no problem. The revised part was colored in red in the revised manuscript and also listed as follows.

"Iron minerals inhibit the growth of bacteria via a free-radical mechanism: Implications for soil carbon storage" (Page 1, Lines 1-2 in the original manuscript)

**was changed to**

"Iron minerals inhibit the growth of *Pseudomonas brassicacearum* J12 via a free-radical mechanism: Implications for soil carbon storage" (Page 1, Lines 1-3 in the revised manuscript)

The reference is listed as follows:

Diaz, J. M., Hansel, C. M., Voelker, B. M., Mendes, C. M., Andeer, P. F., and Zhang, T.: Widespread production of extracellular superoxide by heterotrophic bacteria, Science, 340, 1223-1226, 2013.

- Major comment 2: Integrate your statistical results in the description of the results and in your figures.

**R:** Good comments! In the revised manuscript, we added the statistical results in both the description of the results and figures. The revised parts were colored in red in the revised manuscript and also seen in response to specific comments below.

Specific Comments:

**Abstract**

- "Together, these findings indicate that the reduced surface Fe(II) derived from Fe(III)- containing minerals inhibit bacteria via a free-radical mechanism, which may further contribute to soil carbon storage." : see Major comment 1. Free-radicals may lead to organic matter degradation-mineralization, you do not develop this idea in the manuscript.

**R:** In the revised manuscript, we changed "bacteria" to "J12" and then deleted "which may further contribute to soil carbon storage". The revised part was colored in red in the revised manuscript and also listed as follows.

"Together, these findings indicate that the reduced surface Fe(II) derived from Fe(III)-containing minerals inhibit bacteria via a free-radical mechanism, which may further contribute to soil carbon storage." (Page 2, Lines 33-35 in the original manuscript)

**was changed to**

"Together, these findings indicate that the reduced surface Fe(II) derived from Fe(III)-containing minerals inhibit the growth of *Pseudomonas brassicacearum* J12 via a free-radical mechanism, which may serve as an ubiquitous mechanism between iron minerals and all of the heterotrophic bacteria in view of taxonomically and ecologically diverse heterotrophic bacteria from terrestrial environments as a vast source of superoxide." (Page 2, Lines 34-39 in the revised manuscript)

**Introduction**

- This is a clear introduction which provide a good representation to the overall situation.

**R: Thanks!**

- "The bacterial inhibition property of a mineral is associated with the particular chemistry and with the mineral properties, resulting in the various bacterial inhibition mechanisms of minerals" [1. 43-45]: can you please give more precisions on the various inhibition mechanisms?

**R:** Yes! In the revised manuscript, the various inhibition mechanisms were added. The revised part was colored in red in the revised manuscript and also listed as follows.

"The bacterial inhibition property of a mineral is associated with the particular chemistry and with the mineral properties, resulting in the various bacterial inhibition mechanisms of minerals" (Page 3, Lines 42-45 in the original manuscript)

**was changed to**

"The bacterial inhibition property of a mineral is associated with the particular chemistry and with the mineral properties, resulting in the various bacterial inhibition mechanisms of minerals such as increase of membrane permeability and oxidative damage." (Page 3, Lines 51-54 in the revised manuscript)

- Please, name the Al- and the Fe-contaning minerals that you used in this study, it is hard to understand for non-chemists to which category belong kaolinite, montmorillonite, hematite, goethite, ferrihydrite.

**R:** Good suggestion. In the revised manuscript, we added the explanation of the Aland the Fe-contaning minerals. The added part was colored in red in the revised manuscript and also listed as follows.

"Specifically, montmorillonite and kaolinite are Al(III)-containing minerals, while hematite, goethite and ferrihydrite belong to Fe(III)-containing minerals." (Page 5, Lines 103-104 in the revised manuscript)

**Material and Methods**

- I suggest to separate the paragraphe 2.1 into two parts: "2.1 Mineral preparation" [1.106], "2.2 Pseudomanas cultivation experiments" [1.121]

**R:** In the revised manuscript, we separated the paragraph 2.1 into two parts: "2.1 Mineral preparation", "2.2 *Pseudomanas* cultivation experiments" (**Page 6, Lines 125-140 in the revised manuscript**)

- Suppress "which is a major group of rhizobacteria that aggressively colonize plant

roots, has been considered an important group for sustainable agriculture" [1.122-123]: the information is already given [1.90].

**R:** Agree! In the revised manuscript, this sentence "which is a major group of rhizobacteria that aggressively colonize plant roots, has been considered an important group for sustainable agriculture" was deleted (**Page 6, Line 141 in the revised manuscript**).

- Why didn't you chose to have a control [NB + Mineral]? The OD of this control can be subtracted from the OD measured in [NB + Mineral + Bacteria] and give you the OD of your bacteria without the disturbance induced by the mineral?

**R:** Good comment! In this study, we removed the effect of the OD of mineral on the *Pseudomanas* cultivation experiments based on the protocol of McMahon et al. (2016). In the future, we would like to compare the suggested method by Referee #2 to the protocol of McMahon et al. (2016).

The mentioned reference is listed as follows.

McMahon, S., Anderson, R. P., Saupe, E. E., and Briggs, D. E. G.: Experimental evidence that clay inhibits bacterial decomposers: Implications for preservation of organic fossils, Geology, 44, 867-870, 2016.

- Did you measure the kinetic of bacterial growth during the 12 h? Are you sure that the bacteria is still in exponential phase of growth? Why did you chose 12 h for the first incubation and 8 h for the second one?

**R:** Yes, we had measured the kinetic of bacterial growth and shown as follows. Therefore, we confirm that the bacteria are still in exponential phase of growth from the figure. The protocol we used, i.e., 12 h for the first incubation and 8 h for the second one, is based on the results from McMahon et al. (2016), which is listed in the response of the above question.

Fig. The kinetic of bacterial growth within 15 h (n = 6).

- pH measurement should be explain in "2.6 Chemical analysis"

**R:** Thanks! In the revised manuscript, we added the pH measurement in the section of "2.7 Chemical analysis" and also shown as follows.

"The pH of *Pseudomonas brassicacearum* J12 cultivated with different minerals or without mineral (control) was detected after 12 h." (Page 11, Lines 256-257 in the revised manuscript)

- I do not understand the choice of an ANOVA, when did you used that test.

**R:** One-way ANOVA is a technique that can be used to compare means of two or more samples (using the *F* distribution). Typically, the one-way ANOVA is used to test for differences among at least three groups. When the conditions of normality and homogeneity of variance were met, we considered use this test. In this study, one-sample Kolmogorov-Smirnov Test was used to analyze the distribution of data.

- Figure 1, 3, S5, S6, S7, S8 should integrate your statistical analysis.

**R:** Thanks! We integrated the statistical analysis in the revised manuscript. The revised figures can be seen in the revised manuscript and not listed here for brevity.

- Explain/describe the "one-sample Kolmogorov-Smirnov Test"

**R:** The one sample Kolmogorov-Smirnov test is used to test whether a sample comes from a specific distribution. In this study we used this procedure to determine whether the data set was normally distributed. The added part was colored in red in the revised manuscript and also listed as follows.

"The one sample Kolmogorov-Smirnov test is used to test whether a sample comes from a specific distribution. In this study we used this procedure to determine whether the data set was normally distributed." (Page 11, Lines 265-267 in the revised manuscript)

- Which software did you use to find and represent the model that best fits with your data (Fig.4)?

**R:** In this study, we used SPSS 18.0 to find and represent the model that best fits with our data in Fig.4.

**Results**

- In this part you should not interpret your results: [1.230-231], [1.244-247], [1.294-295], [1.268-269], [1.280-281], [1.313], [1.318-319], [1.327-328].

**R:** Agree! In the revised manuscript, we deleted these parts that interpret the results. The revised figures could be seen in the revised manuscript and not listed here for brevity.

- Here, we are waiting for an exhaustive description of the results obtained during the study: give some values (mean  $\pm$  SE) and precise when values are significantly (or not) different between the different conditions tested.

**R:** Great comment! We integrated the description of the results and the statistical analysis in the revised manuscript. The revised parts could be seen in the revised manuscript and not listed here for brevity.

Paragraph [1.224-234]:

- "3.1. Bacterial inhibition by minerals" [1.223]: This title does not correspond in case of montmorillonite. I suggest something like: "Effect of mineral nature and their concentrations on P. brassicacearum J12 development".

**R:** Good suggestion! The revised part was colored in the revised manuscript and also listed as follows.

"3.1. Effect of mineral nature and their concentrations on J12 development " (Page 12, Line 273 in the revised manuscript).

- Suppress "The effects of the nature and content of tested minerals on the OD 600 of Pseudomonas brassicacearum J12 subcultures taken after 12 h growth are shown in Fig. 1.": it should be in the "Material and Methods" part.

**R**: Agree! In the revised manuscript, we deleted this sentence. (**Page 12, Line 274 in the revised manuscript**)

- "Compared to Control (i.e., no minerals), the presence of montmorillonite significantly increased OD 600.": give values.

**R**: We added the values in the revised manuscript. The revised part was colored in the revised manuscript and also listed as follows.

"Compared to Control (0.34  $\pm$  0.01), the presence of montmorillonite significantly (p < 0.05) increased OD600 (Fig. 1). Specifically, the OD600 values of samples were 0.43  $\pm$  0.01, 0.44  $\pm$  0.02 and 0.43  $\pm$  0.01 at the concentration of 5, 10 and 25 mg mL-1, respectively." (**Page 12, Lines 274-277 in the revised manuscript**)

- Suppress "On the other hand," [1. 227]

**R: Done.**

- [1.227-230]: "Presence of all other investigated minerals decreased OD 600 in the following order: ferrihydrite > goethite > hematite > kaolinite at 5 and 25 mg mL-1, and ferrihydrite > goethite > kaolinite > hematite at 10 mg mL-1": Please give some values.

**R:** Done. The revised part was colored in the revised manuscript and also listed as follows.

"Presence of all other investigated minerals decreased  $OD_{600}$  in the following order: ferrihydrite (0.24 ± 0.04 and 0.09 ± 0.01) > goethite (0.26 ± 0.02 and 0.14 ± 0.00) > hematite (0.30 ± 0.03 and 0.16 ± 0.02) > kaolinite (0.32 ± 0.01 and 0.20 ± 0.01) at 5 and 25 mg mL-1, respectively, and ferrihydrite (0.16 ± 0.02) > goethite (0.18 ± 0.02) > kaolinite (0.21 ± 0.02) > hematite (0.28 ± 0.02) at 10 mg mL-1. An increase in mineral concentration resulted in a significant decrease in  $OD_{600}$ ." (Page 12, Lines 277-283 in the revised manuscript)

- Suppress "Meanwhile" [1.232]

**R: Done.**

- "An increase in mineral concentration resulted in a significant decrease in OD 600, except for montmorillonite" [l. 232-233]: Give some values.

**R:** We added the values and shown as follows.**

"Compared to Control  $(0.34 \pm 0.01)$ , the presence of montmorillonite

significantly ( $p \le 0.05$ ) increased OD600 (Fig. 1)" (**Page 12, Lines 274-279 in the revised manuscript**)

"An increase in mineral concentration resulted in a significant (p < 0.05) decrease in OD600. However, in presence of montmorillonite the OD600 is stable at about 0.43 for all the mineral concentration studied." (**Page 12, Lines 282-284 in the revised manuscript**)

- Suppress "as the OD 600 seemed to be independent of its concentration" [1.233-234]: it is an interpretation. You can replace it by something like: "However, in presence of montmorillonite the OD600 is stable at  $0.43 \pm SE$  for all the mineral concentration studied"

**R:** Agree! We revised "as the OD 600 seemed to be independent of its concentration" and shown as follows.

"except for montmorillonite, as the  $OD_{600}$  seemed to be independent of its concentration" (Page 12, Lines 233-234 in the original manuscript)

was changed to

"However, in presence of montmorillonite the OD600 is stable at about 0.43 for all the mineral concentration studied" (Page 12, Lines 283-284 in the revised manuscript)

- Fig.1: we do not see bottom bar of the SE

**R**: In the revised manuscript, we added the bottom bar of the SE.

- Fig.1 text/description: you should mention the mineral concentrations.

**R:** In the revised manuscript, we added the description of the mineral concentrations and shown as follows.

"Gray, magenta and cyan represent the mineral concentration of 5, 10 and 25 mg mL-1, respectively." (**Page 35, Lines 1094-1096 in the revised manuscript**)

Paragraph [1.235-247]:

- It represents a new idea: you should give it a title (e.g. "chemical structure of minerals")

**R:** Agree! In the revised manuscript, we added the title as "3.2. Chemical structure of minerals". (**Page 12, Line 285 in the revised manuscript**)

- [1.235-247]: why didn't you describe the EPR profiles of ferrihydrite, goethite and hematite?

**R:** Electron paramagnetic resonance (EPR) spectroscopy is a method for studying materials with unpaired electrons. Therefore, it cannot be applied to examine paramagnetic substances, e.g. iron oxides. In this study, we cannot describe the EPR profiles of iron oxides.

Paragraph [1.249-258]:

- "A 12 h cultivation of Pseudomonas brassicacearum J12 in the presence of different minerals revealed that generation of HO' radicals in the cases of montmorillonite, kaolinite and hematite was almost similar to the control (Fig. 3)": "Almost"? You should precise if the difference are significant or not. To precise my comment, you should study the significance of the difference between the control and montmorillonite for the three concentrations, and kaolinite 25 mg mL-1. Moreover, I think that the difference is significant between (i) montmorillonite 25 mg mL-1 and kaolinite 25 mg mL-1 and (ii) montmorillonite 25 mg mL-1.

**R:** Agree! In the revised manuscript, we revised this sentence and also listed as follows.

" A 12 h cultivation of J12 in the presence of different minerals revealed that generation of HO' radicals in the cases of montmorillonite, kaolinite and hematite was similar (p > 0.05) to the control at low concentration (i.e., 5 mg mL-1) but significant different (p < 0.05) at high concentration (i.e., 25 mg mL-1) (Fig. 3)." (Page 13, Lines 357-360 in the revised manuscript)

- [1.255]: replace "rapidly" by significantly: there is no notion of time.

**R: Done.**

- Fig.3 text/description: you should mention the mineral concentrations

**R:** Done. The revised parts were colored in red in the revised manuscript and also listed as follows.

"Figure 3. Generation of hydroxyl radical (HO') after 12 h growth of *Pseudomonas brassicacearum* J12 with different minerals and with no minerals (control). Al-containing minerals: K, kaolinite; M, montmorillonite. Fe-containing minerals: H, hematite; G, goethite; F, ferrihydrite. C, Control (i.e., no mineral). Gray, magenta and cyan represent the mineral concentration of 5, 10 and 25 mg mL-1, respectively. Values are the mean  $\pm$  SE (n = 3)." (Page 37, Lines 1105-1110 in the revised manuscript)

"A 12 h cultivation of J12 in the presence of different minerals revealed that 18

generation of HO• radicals in the cases of montmorillonite, kaolinite and hematite was similar (p > 0.05) to the control at low concentration (i.e., 5 mg mL-1) but significant different (p < 0.05) at high concentration (i.e., 25 mg mL-1) (Fig. 3). However, presence of goethite and ferrihydrite significantly increased the production of HO• radicals, which increased with an increase in their concentration. Specifically, in ferrihydrite treatments, the concentration of HO• was approximately 260 nM at 5 and 10 mg mL-1 but increased significantly to 450 nM at 25 mg mL-1. In addition, the generation of HO• at early growth (i.e., 2 h) was only detected with ferrihydrite at both 10 and 25 mg mL-1 and with goethite at 25 mg mL-1 (Fig. S6)." (**Page 13, Lines 357-366 in the revised manuscript**)

Paragraph [1.259-295]:

- Globally, I encourage the authors to reorganize this part of the manuscript. You should describe all your results (Fig. 4) not only those which are consistent with your interpretation. Just for example: I observe a significant increase of soluble Fe in the treatment containing goethite with the increase of goethite concentration but this results is missing from the test.

**R:** Thanks! In the revised manuscript, we had reorganized this part and also listed as follows.

"To explore the factors affecting the generation of HO• and the inhibition of J12, we examined iron chemistry and its correlation with HO $\circ$  and OD600 (Fig. 5). Much more soluble Fe at 12 h was released from Fe(III)-containing minerals (6.7-27, 21-36 and 41-107 mg L-1 for hematite, goethite and ferrihydrite, respectively) than from montmorillonite (~0.3 mg L-1), kaolinite (~0.6 mg L-1), and control (~0.4 mg L-1) (Fig. 5a). With the increase of concentration, soluble Fe significantly (p < 0.05) increased at both 2 h and 12 h for ferrihydrite, only at 12 h for goethite. As for hematite, significant (p < 0.05) increase was only observed from 5 to 10 mg L-1 at 12 h (Fig. S7). The solubility of Fe was closely related to redox potential and pH value (Fig. S8). Results showed that Eh of bacteria-mineral mixture after incubation was generally lower than the suspension of minerals alone (Table S5), suggesting that the redox potential was decreased by the interaction between mineral and J12. Furthermore, the solution pH was determined after 12 h growth of J12 with different minerals and with no minerals (control) (Fig. 4). The range of solution pH varied from 4 to 6 for all of the treatments, except for ferrihydrite treatment with a pH near 7. For all of the examined minerals, the trends at 12 h were similar in the following order (total Fe and Fe(II)): ferrihydrite (760-3588 and 182-488 mg  $L^{-1}$ ) >> goethite  $(48-127 \text{ and } 31-94 \text{ mg } \text{L}^{-1}) > \text{hematite } (15-82 \text{ and } 9-35 \text{ mg } \text{L}^{-1}) > \text{montmorillonite}$ (5-10 and 4-8 mg  $L^{-1}$ ), kaolinite (10-12 and 4-9 mg  $L^{-1}$ ) or control (7 and 6 mg  $L^{-1}$ ) (Fig. 5b-5c). A significant difference of total Fe in solutions containing 25 mg mL-1 ferrihydrite between 2 h and 12 h may be attributable to the aging of a portion of ferrihydrite to its more crystalline counterparts, as revealed by micro X-ray fluorescence (µ-XRF). The more crystalline counterparts could not be dissolved by the modified 1,10-phenanthroline method.

Furthermore, a positive correlation exists between HO' and soluble Fe content (R

= 0.92, t = -3.49, p = 0.003) and Fe(II) (R = 0.98, t = -4.28, p = 0.001) (Fig. 5d and 5f, Table S2). However, a significant but negative correlation between  $OD_{600}$  and soluble Fe (R = -0.57, t = 2.99, p = 0.009), and Fe(II) (R = -0.81, t = 2.23, p = 0.038) was found (Fig. 5g and 5i). Moreover, the correlation between HO' and Fe(III) (R = 0.94, t = 1.38, p = 0.19), and between OD600 and Fe(III) (R = -0.80, t = 1.67, p = 0.116) were not significant (Fig. 5e and 5h). To test whether the release of Fe(III) to solution inhibit the growth of J12 via a free-radical mechanism, we replaced Fe(III)-containing minerals by adding a series of concentrations of  $Fe(NO_3)_3$ , i.e., 0, 50 and 100 mg L- in the cultivation experiments with the final pH of 7.2. The results showed that addition of Fe(III) can inhibit the growth of J12 (25-50%) by producing an additional HO' concentration of 15 nM (Fig. S9), supporting the role of Fe(III) ion from solution in the initialization of a free-radical reaction. In addition, the inhibition of soluble Fe on J12 was more important in the concentration of 100 mg  $L^{-1}$  than that of 50 mg  $L^{-1}$ while HO• production still kept the same between those two concentrations (Fig. S9). The reason of this phenomenon may attributable to the intracellular oxidative damage of soluble Fe that penetrated into cells and triggering of intracellular ROS generation.

In addition, we also examined soluble Al during the cultivation experiments (Fig. 6a) and found a high concentration of Al in the montmorillonite and kaolinite solutions. However, almost no correlation was found between soluble Al and HO• (R = -0.35, t = -3.36, p = 0.004) and OD600 (R = 0.30, t = 2.24, p = 0.041) (Fig. 6b-6c)." (Pages 13-15, Lines 369-482 in the revised manuscript)

- Given the importance of the pH in the results description, I think that this result may be integrated in Fig.4.

**R**: Agree! In the revised manuscript, pH results was added as Fig.4.

- [l. 261-263]: "Much more soluble Fe was released from Fe(III)-containing minerals than from montmorillonite, kaolinite, and control (Fig. 4a)": Please give some values.

**R:** Done. The revised part was colored in red in the revised manuscript and also listed as follows.

"Much more soluble Fe at 12 h was released from Fe(III)-containing minerals (6.7-27, 21-36 and 41-107 mg  $L^{-1}$  for hematite, goethite and ferrihydrite, respectively) than from montmorillonite (~0.3 mg  $L^{-1}$ ), kaolinite (~0.6 mg  $L^{-1}$ ), and control (~0.4 mg  $L^{-1}$ ) (Fig. 5a). " (Page 13, Lines 369-373 in the revised manuscript)

- "The solubility of Fe is closely related to pH value.": Are you sure about that? The pH of goethite solution is equivalent to the pH of kaolinite, montmorillonite, hematite and goethite but the solubility of Fe in solution containing hematite and goethite seems to be more important. You should draw the graph showing the correlation between pH and the soluble Fe.

**R:** Yes. In the revised manuscript, we draw the graph showing the correlation between pH and the soluble Fe and listed as Fig. S8.

Figure S8. Correlation between pH and the soluble Fe.

- Fig.4.b: You have a surprising result: the significant decrease of Total Fe in solutions containing 25 mg mL-1 ferrihydrite between 2 h and 12 h. How do you explain that?

**R:** In this study, total Fe was determined by a modified 1,10-phenanthroline method (Amonette, 1998). This method dissolved Fe by HCl. Therefore, this method may not enough to extract all of Fe from crystalline Fe minerals (e.g. hematite and goethite) that were detected in ferrihydrite samples after 12 h cultivation (Fig. 7 and Table S3). Therefore, we inferred that the aging of a portion of ferrihydrite to its more crystalline counterparts may be the possible reason about the significant decrease of Total Fe in solutions containing 25 mg mL-1 ferrihydrite between 2 h and 12 h. In the revised manuscript, the corresponding explanation was added and also listed as follows.

"A significant difference of total Fe in solutions containing 25 mg mL-1 ferrihydrite between 2 h and 12 h may be attributable to the aging of a portion of ferrihydrite to its more crystalline counterparts, as revealed by  $\mu$ -XRF, which could not be dissolved by the modified 1,10-phenanthroline method." (Page 14, Lines 409-413 in the revised manuscript)

- "For all of the examined minerals, the trends of total Fe and Fe(II) were similar in the following order: ferrihydrite » goethite > hematite > montmorillonite  $\approx$  kaolinite  $\approx$  control (Fig. 4b-4c)": Please give some values. What do you mean with » and  $\approx$ ? Is there a relation with a statistical analysis?

**R:** In the original manuscript, ">" represents the former being far greater than the latter, while " $\approx$ " indicates the former being a close to the latter. In the revised manuscript, we added the values and shown as follows.

"For all of the examined minerals, the trends of total Fe and Fe(II) were similar in the following order: ferrihydrite > goethite > hematite > montmorillonite  $\approx$

kaolinite  $\approx$  control (Fig. 4b-4c)" (Page 13, Lines 264-268 in the original manuscript)

**was changed to**

"For all of the examined minerals, the trends at 12 h were similar in the following order (total Fe and Fe(II)): ferrihydrite (760-3588 and 182-488 mg  $L^{-1}$ ) > goethite (48-127 and 31-94 mg  $L^{-1}$ ) > hematite (15-82 and 9-35 mg  $L^{-1}$ ) > montmorillonite (5-10 and 4-8 mg  $L^{-1}$ ), kaolinite (10-12 and 4-9 mg  $L^{-1}$ ) or control (7 and 6 mg  $L^{-1}$ ) (Fig. 5b-5c)." (Page 14, Lines 405-409 in the original manuscript)

- [1.274-275] "Furthermore, a positive correlation exists between OD600 and soluble Fe content (R = 0.92, t = -3.49, p = 0.003) and Fe(II) (R = 0.98, t = -4.28, p = 0.001) (Fig. 4d and 4f, Table S2).": I think that you wanted to say "a positive correlation between Hydroxyl radical content and soluble Fe content".

**R:** Yes! In the revised manuscript, we changed  $OD_{600}$  to HO' and listed as follows.

"Furthermore, a positive correlation exists between OD600 and soluble Fe content (R = 0.92, t = -3.49, p = 0.003) and Fe(II) (R = 0.98, t = -4.28, p = 0.001) (Fig. 4d and 4f, Table S2)." (Page 13, Lines 274-276 in the original manuscript)

**was changed to**

"Furthermore, a positive correlation exists between HO• and soluble Fe content (R = 0.92, t = -3.49, p = 0.003) and Fe(II) (R = 0.98, t = -4.28, p = 0.001) (Fig. 5d and 5f, Table S2)." (Page 14, Lines 414-416 in the revised manuscript)

- The interpretation of "R" and "t" should appear in the Material and Methods.

**R:** Thanks! In the revised manuscript, the interpretation of parameters was added and also listed as follows.

"In the regression equation, the parameters *R* and *t* represent coefficient of determination and t-test." (Pages 11-12, Lines 267-269 in the revised manuscript)

- [1.277]: "R=-0.75" and "t= 2.27" do not correspond to the values in the Fig.4.

**R:** Thanks! We revised the values and shown as follows.

"However, a significant but negative correlation between  $OD_{600}$  and soluble Fe (R = -0.75, t = 2.99, p = 0.009), and Fe(II) (R = -0.81, t = 2.27, p = 0.038) was found (Fig. 4g and 4i)." (**Page 13, Lines 276-279 in the original manuscript**)

**was changed to**

"However, a significant but negative correlation between  $OD_{600}$  and soluble Fe (R = -0.57, t = 2.99, p = 0.009), and Fe(II) (R = -0.81, t = 2.23, p = 0.038) was found (Fig. 5g and 5i)." (Page 14, Lines 416-418 in the revised manuscript)

- Fig.S7 should appear in Fig.4

**R:** The results of Fig.S7 were derived from an independent experiment, which was totally different from those of Fig. 4. Therefore, we did not combined Fig.S7 into Fig.4.

- Fig.S7: Can you explain why the inhibition of Pseudomonas is more important in Fe(III) 100mg  $L^{-1}$  than in Fe(III) 50mg  $L^{-1}$  while hydroxyl radical production still the same between those two concentrations? Is that not the sign of the existence of another process involved in the Pseudomonas growth inhibition? I find this result very important, it should be considered in your discussion.

**R:** In this study, HO• trapped by TPA is mainly extracellular. Structural Fe(II), not soluble Fe2+, was responsible for extracellular HO• production. However, the toxicity of soluble Fe may also contribute to the observed cell killing by penetrating into cells and triggering of intracellular ROS generation (Williams et al., 2011). The reason of this phenomenon may due to the intracellular oxidative damage of Fe. Consistent with the recent study (Wang et al., 2017), inhibition activity of Fe minerals is a result of followed two factors: (1) HO• production extracellularly from structural Fe(II); (2) intracellularly from soluble Fe. In the revised manuscript, we added the related explanation about Fig. S7 and also listed as follows.

"In addition, the inhibition of soluble Fe on J12 was more important in the concentration of 100 mg L-1 than that of 50 mg L-1 while HO• production still kept the same between those two concentrations (Fig. S9). The reason of this phenomenon may attributable to the intracellular oxidative damage of soluble Fe that penetrated into cells and triggering of intracellular ROS generation." (Page 15, Lines 474-478 in the revised manuscript)

Paragraph [1.296-328]:

- For non-chemist, this part is difficult to understand. Maybe the next comments will allow you to make it more accessible for biologists.

**R:** Thanks! We try to explain it more accessible.

- Fig.5.a: What do the colors mean?

**R:** Colors in Fig.5a represent the different density of Fe in the selected area. Red color

represents high density of Fe, followed by orange, yellow, green, little green, and purple. In the revised manuscript, we added the explanation of color in the caption of Fig. 5. The added part was colored in red in the revised manuscript and also listed as follows.

"Figure 5. Correlative micro X-ray fluorescence ( $\mu$ -XRF) and synchrotron-based Fourier transform infrared (SR-FTIR) analysis of the thin section from the cultures of the 25 mg/mL ferrihydrite treatment after 12 h cultivation. (a)  $\mu$ -XRF map. (b) The LCF fitting of  $\mu$ -X-ray absorption near-edge structure (XANES) analysis the selected regions of interest (ROI) region (i.e., A and B). (c) SR-FTIR maps. The color scale in (c) is a relative scale for each peak height and does not allow quantitative comparisons between peaks." (Page 37, Lines 689-695 in the original manuscript)

**was changed to**

"Figure 7. Correlative micro X-ray fluorescence ( $\mu$ -XRF) and synchrotron-based Fourier transform infrared (SR-FTIR) analysis of the thin section from the cultures of the 25 mg/mL ferrihydrite treatment after 12 h cultivation. (a)  $\mu$ -XRF map. (b) The LCF fitting of  $\mu$ -X-ray absorption near-edge structure (XANES) analysis the selected regions of interest (ROI) region (i.e., A and B). (c) SR-FTIR maps. Red color in (a) represents high density of Fe, followed by orange, yellow, green, little green, and purple. Red color in (c) indicates the highest intensity of functional groups, followed by yellow, green, and blue. The color scale in (c) is a relative scale for each peak height and does not allow quantitative comparisons between peaks." (Page 41, Lines 1138-1146 in the revised manuscript)

- Why did you select those regions of the spectra for XANES?

**R:** Two spots represent the internal and external of the selected particles, respectively. We want to observe the changes of Fe species from outside to inside, thus the spots were selected for XANES analysis.

- Fig.5.c: What do the colors mean?

**R:** Blue represents the background (i.e., the intensity nears zero), while red color represents the highest intensity of functional groups, followed by yellow and green. In the revised manuscript, we added the explanation of color in the caption of Fig. 5-c. The added part was colored in red in the revised manuscript and also listed as follows.

"**Figure 7.** Correlative micro X-ray fluorescence ( $\mu$ -XRF) and synchrotron-based Fourier transform infrared (SR-FTIR) analysis of the thin section from the cultures of the 25 mg/mL ferrihydrite treatment after 12 h cultivation. (a)  $\mu$ -XRF map. (b) The LCF fitting of  $\mu$ -X-ray absorption near-edge structure (XANES) analysis the selected regions of interest (ROI) region (i.e., A and B). (c) SR-FTIR maps. Red color in (a) represents high density of Fe, followed by orange, yellow, green, little green, and purple. Red color in (c) indicates the highest intensity of functional groups, followed by yellow, green, and blue. The color scale in (c) is a relative scale for each peak height and does not allow quantitative comparisons between peaks." (Page 41, Lines 1138-1146 in the revised manuscript)

- Conserve the same colors between Fig.5.b Spot A and B.

**R:** Thanks! We revised the colors in Fig.5-b.

- [1.309]: Spot A or Spot B?

**R:** Thanks! We revised them in the revised manuscript and also listed as follows.

"with a lesser percentage (~17%) of  $\text{FeC}_2\text{O}_4$  among the mineral particles (Spot A in Fig. 7b and Table S3). However, considerable percentages of hematite (~13%), goethite (~19%) and  $\text{FeC}_2\text{O}_4$  (~25.9%) were present on the edge of these mineral particles (Spot B in Fig. 7b and Table S3)." (Pages 15-16, Lines 493-515 in the revised manuscript)

- [1.307-309]: why don't you speak about  $FeC_2O_4$  (25.9%) in spot B?

**R:** We added the  $FeC_2O_4$  (~25.9%) in the revised manuscript and also listed as follows.

"considerable percentages of hematite ( $\sim$ 13%) and goethite ( $\sim$ 19%) were present on the edge of these mineral particles (Spot B in Fig. 5b and Table S3)." (**Page 14, Lines 308-309 in the original manuscript**)

was changed to

"considerable percentages of hematite (~13%) , goethite (~19%) and FeC2O4 (~25.9%) were present on the edge of these mineral particles (Spot B in Fig. 7b and Table S3)." (Page 16, Lines 513-515 in the revised manuscript)

- Why is there goethite and hematite in sample which only contain ferrihydrite?

**R:** During the incubation, a portion of ferrihydrite will transform to its more crystalline counterparts, such as hematite and goethite, by J12, owing to the so-called "aging" process.

- You should give different title to the paragraph [1.298-309], [1.310-320]

**R:** Agree. In the revised manuscript, a different title was added and also listed as

follows.

**"3.6.** Effect of the presence of J12 on surface Fe species" (Page 16, Line 523 in the revised manuscript)**

- Paragraph [1.321-328] should be describe in paragraph [1.298-309]: it is the same figure and consequently the same idea.

**R: Done.**

- [1.317-320] "Interestingly, the area of the peak at 709.5 eV was bigger in the F + bacteria treatment than that in F - bacteria treatment (Fig. 6b-6c), suggesting that Fe(II) was generated on the surface of ferrihydrite during the cultivation with bacteria. Based on the reaction 1, HO' should be the oxidant products.": Is that reproducible between samples? Is that spectrum the mean representation of several spectra?

**R**: No. The two spectra were not reproducible but ferrihydrite cultivated with (F + bacteria) and without (F - bacteria) bacteria. That spectrum was measured once rather than mean one of several spectra. However, the spectra should be representative, owing to the prepared samples for XPS measurement being uniform.

- Fig.6 b and c: what do the colors mean?

**R:** Dark line represents the raw spectrum, orange line represents the fitted spectrum, and other lines represent the component of fitted Fe species. The above explanation was added in the revised manuscript and also listed as follows.

"Figure 8. (a) Fe 2p X-ray photoelectron spectroscopy (XPS) spectra of ferrihydrite samples, F+bacteria and F-bacteria; (b-c) Fe 2p 3/2 spectra of F+bacteria and F-bacteria, respectively, during the cultivation (12 h). F+bacteria, ferrihydrite with bacteria; F-bacteria, ferrihydrite without bacteria. In subfigure (b) and (c), dark, orange and other lines represents the raw spectrum, fitted spectrum and the component of fitted Fe species." (Page 42, Lines 1151-1156 in the revised manuscript)

- Fig.6 a: Correct "Inyensity" by "Intensity"

**R: Done.**

- [1.325]: "good", can you precise this term please?

**R:** Thanks! We replaced "good" with "significant" in the revised manuscript.

New paragraph:

- Given the importance of Al in your discussion (half of the discussion), the Fig.S8 should appear in the main manuscript (not in Sup Mat) with the results presented in Fig.4.

**R:** Agree! In the revised manuscript, we moved Fig.S8 to the main manuscript as Fig.6.

- In Fig.1, Fig.3, Fig.4, Fig.S6, Fig.S8: you should distinguish Al- from Fe-containing minerals.

**R:** Agree! We added the description of Al- from Fe- containing minerals in the in the revised manuscript and distinguish them in the captions, which could be seen in the revised manuscript.

Discussion

Another time: I disapprove the use of the term "bacteria" which may refer to the domain of bacteria (see main comment 1).

**R:** As the response to Comment 1, we changed "bacteria" to more specific "J12" throughout the whole manuscript.

4.1. Effect of Al(III)-containing minerals on the inhibition of bacterial growth

- [1.336-343]: "It should be noted that the presence of minerals may potentially interfere with the measurement of cell numbers in Fig. 1. In this study, we subsampled the experimental cultures and diluted them in fresh medium so that both clay particles and bacteria were 200× less concentrated (Fig. S3), following the protocol of McMahon et al. (2016). As a result, the effect of mineral concentration may be minimal. In addition, plating the bacteria by evaluating populations by counting colonies may act as a complementary method for OD600 and needs to be investigated in the future.": I am not waiting for a response to that comment: In your case, I would have chosen the association of a cell labeling with DAPI and a count of labeled cells with flow cytometry (or fluorescence microscopy).

**R:** Thanks! In the revised manuscript, we added the association of a cell labeling with DAPI and a count of labeled cells with flow cytometry (or fluorescence microscopy) as an alternative choose. The added part was colored in red in the revised manuscript and also listed as follows.

"Furthermore, the association of a cell labeling with DAPI and a count of labeled cells with flow cytometry (or fluorescence microscopy) is also an alternative choose." (Page 17, Lines 586-588 in the revised manuscript)

- [1.355-357]: "Furthermore, the formation of some Al intermediates by the decreasing pH, such as  $Al_{13}O_4(OH)_{24}^{7+}$ , is also suggested to be more toxic for bacterial growth (Amonette et al., 2003; Liu et al., 2016).": what pH are you referring to? Is that in

**accordance with the pH measured in your study?**

**R:** Good comment! The pH was referred to the solution pH. In this study, we did not detect a significant decrease of pH (see Fig. 4 in the revised manuscript). Therefore, we added the corresponding discussion in the revised manuscript and also listed as follows.

"However, we did not detect a significant decrease of pH in this study (Fig. 4), suggesting that the formation of some Al intermediates may be slightly." (Page 18, Lines 617-619 in the revised manuscript)

- The information given at [1.357-359] should appear after [1.345-349]. Then, you can discuss (i) on the results that you expected to observe and (ii) on the interpretation of the results that you obtained.

**R:** Agree! Done.**

4.2. Inhibition of bacteria by Fe(III)-containing minerals via a free-radical mechanism

- If we take the two equations cited in your introduction: (1)  $\equiv$ Fe(III)-OH + H2O2  $\rightarrow$  Fe(II) + H2O + HO2 (2)  $\equiv$ Fe(II) + H2O2  $\rightarrow$  Fe(III)-OH + HO• Where does H2O2 come from? Pseudomonas? If it come from the bacteria, the reduction of it development should induce a decrease of HO• production in LB medium containing Fe minerals (if H2O2 is the limiting compound in the reaction, and it should be the case here), am I wrong?

**R:** Yes. *Pseudomonas* J12 could produce  $H_2O_2$ . The reduction of  $H_2O_2$  along with the oxidation of Fe induce an increase of HO• based on the following equation:

 ${}_{\equiv}Fe(II) + H_2O_2 \rightarrow Fe(III)\text{-}OH + HO^{\bullet}$

- Correct the sentence [1.383-387]: "In line with other studies (Kwan and Voelker, 2003; Wang et al., 2017b), we deduced that HO' may mainly generate on the mineral surface, partly due to the positive charge of mineral surface (Tombácz and Szekeres, 2006) but the negative charge of microbes (Jucket et al., 1996)."

**R:** Thanks! This sentence was changed to "In our experiment, there was a lesser amount of HO• produced with the different concentrations of aqueous  $Fe(NO_3)_3$  (Fig. S9) than with the iron minerals (Fig. 3), which was in line with other studies (Kwan and Voelker, 2003; Wang et al., 2017b). Therefore, we deduced that HO• may mainly generate on the mineral surface, partly due to the positive charge of mineral surface (Tombácz and Szekeres, 2006) but the negative charge of microbes (Jucket et al., 1996)." (Page 19, Lines 666-672 in the revised manuscript)

- [1.401-404]: "High percentage of the less stable ferrihydrite (Table S3) may be

attributable to the stabilization role of produced EPS (Fig. 5c) by bacteria to minerals, which had been shown during the cultivation of fungi with minerals (Li et al., 2016). Please, divide this sentence into two sentences in order to distinguish your contribution from the contribution of Li et al. (2016). Can you precise your idea on the role of EPS on stabilization process please?

**R:** Agree ! We divided this sentence into two sentences and shown as follows.

"High percentage of the less stable ferrihydrite (Table S3) may be attributable to the stabilization role of produced EPS (Fig. 5c) by bacteria to minerals, which had been shown during the cultivation of fungi with minerals (Li et al., 2016)." (Page 18, Lines 401-404 in the original manuscript)

**was changed to**

"High percentage of the less stable ferrihydrite (Table S3) may be attributable to the stabilization role of produced EPS (Fig. 5c) by J12 to minerals. It is consistent with a previous finding in the cultivation of fungi with minerals (Li et al., 2016). The stabilization role of EPS was mainly identified as its combination into the network structure of minerals, which prevents the formation of crystalline minerals (Braunschweig et al., 2013)" (Page 20, Lines 695-701 in the revised manuscript)

Braunschweig J., Bosch J., and Meckenstock R. U.: Iron oxide nanoparticles in geomicrobiology: from biogeochemistry to bioremediation, New Biotechnol., 30, 793-802, 2013.

- [1.414]: suppress "cellular". I do not understand the difference between cellular and free reductant? Free reductant such as FADH2 are intracellular, no? I think that you want to separate (i) cellular from (ii) non-cellular reactions, am I wrong?

**R:** We agree with the comment! We revised the sentence and shown as follows.

"In addition to Fenton-like reactions (Garrido-Ramírez et al., 2010), Fe(II) can also be generated by catalyzing a series of cellular intracellular (e.g., glutathione and NAD(P)H) and free (e.g., L-cysteine and FADH2) reductants (Imlay, 2003)." (**Page 19**, **Lines 413-415 in the original manuscript**)

was changed to

"In addition to Fenton-like reactions (Garrido-Ramírez et al., 2010), Fe(II) can also be generated by catalyzing a series of intracellular reductants (e.g., glutathione, NAD(P)H, L-cysteine and FADH2) (Imlay, 2003)." (**Page 20, Lines 710-712 in the revised manuscript**).

- [1.413-418]: Are those reactions linked to HO production?

**R**: Yes. These reactions promote the formation of Fe(II) which reacts with  $H_2O_2$  through Fenton reactions that can accelerate the generation of HO•.

- [1.424]: Given that results in Fig.1 and Fig.S7 are produce by different experimental device, are you sure that you can give this interpretation to your results?

**R:** Yes. According to the results of soluble Fe, the concentrations were about 0-100 mg/L. In order to observing the effects of soluble Fe on the inhibition of bacteria, the concentrations of soluble Fe were set as 0, 50, 100 mg/L.

- [1.427]: "simultaneously"?

**R:** In the revised manuscript, we changed "simultaneously" with "also" and the revised sentence was also listed as follows.

"Intracellular oxidative toxicity also caused by soluble Fe(III) played an important role in the inhibition activity (Schoonen et al., 2006)" (Page 21, Lines 735-737 in the revised manuscript).

4.3. Inhibition of bacterial growth by a free-radical mechanism and its implications for soil carbon storage

- Fig.7: Can you please explain the figure in the caption?

**R:** Yes. In the revised manuscript, we changed "simultaneously" with "also" and the revised sentence was also listed as follows.

"Figure 9. Schematic of the heterotrophic bacterial inhibition by Fe(III)-containing minerals through a free-radical mechanism. Reactions 1-4 represent the processes occurring at heterotrophic bacteria-mineral interfaces and are detailed in the main text. ① Production of HO• through the Fenton or Fenton-like reactions; ② Direct inhibition of heterotrophic bacteria by HO•; ③ Indirect inhibition of heterotrophic bacteria by HO•; ④ Intracellular inhibition of heterotrophic bacteria by HO•; ④ Intracellular inhibition of heterotrophic bacteria by Soluble Fe." (Page 43, Lines 1160-1165 in the revised manuscript).

- Can you please go further in the processes through which soluble  $Fe^{3+}$  and  $Fe^{2+}$  will have an inhibition effect on Pseudomonas?

**R:** Yes. We added the description in the revised manuscript and shown as follows.

"Soluble Fe(II) and Fe(III) released from minerals can penetrate into the cell

membranes, thereby inducing intracellular oxidative damage (Williams et al., 2011)." (Page 21, Lines 746-748 in the revised manuscript)

- Can you please go further in processes through which HO• will have a "direct" inhibition power on Pseudomonas (modification of cell membrane physico-chemical properties?)

**R**: Yes. HO• can modify the cell membrane physico-chemical properties. We added the description in the revised manuscript and shown as follows.

"Oxidative damage of extracellular HO' may lead to bacterial inactivation, and protection of carbon from microbial degradation." (Page 20, Lines 436-437 in the original manuscript)

**was changed to**

"Oxidative damage of HO• may induce the damage of a membrane lipid and cardiolipin that can lead to heterotrophic bacterial inactivation (Wang et al., 2017). In soil, heterotrophic bacteria are the main driver of soil carbon decomposition and greenhouse gas emission. As a result, the inactivation of heterotrophic bacteria results in protection of carbon from microbial degradation." (Page 21, Lines 748-752 in the revised manuscript)

- [1.436-437]: "Oxidative damage of extracellular HO• may lead to bacterial inactivation, and protection of carbon from microbial degradation." Please go further in this interpretation: HO• have a role on Pseudomonas growth (it is your study), but HO• can have other impacts in soils. What are they? How can HO• and Fe act (i) on the soil C storage and (ii) on the soil C degradation-mineralization?

**R:** Oxidative damage of HO $\cdot$  may induce the damage of a membrane lipid and cardiolipin that can lead to heterotrophic bacterial inactivation. In soil, heterotrophic bacteria are the main driver of soil carbon decomposition and greenhouse gas emission. As a result, the inactivation of heterotrophic bacteria results in protection of carbon from microbial degradation.

HO• do have other impacts in soils. Except for decomposition of soil organic carbon (SOC), the presence of HO• can also stabilize C in soil via a rapid formation of new intermolecular covalent bonds among soil components (Piccolo *et al.*, 2011). In addition, the mobilized Fe can be easily transformed into the newly-formed reactive Fe (hydro)oxides (especially poorly crystalline Fe oxides) (Kleber et al., 2005; Yu et al., 2017), which will promote the formation of organo-mineral associations that are chemically more stable (Koegel-Knabner et al., 2008).

In the revised manuscript, we added the interpretation and discussion about the effect of HO• and Fe on the soil C storage. The revised parts were colored in red in the

revised manuscript and also listed as follows.

"Oxidative damage of HO• may induce the damage of a membrane lipid and cardiolipin that can lead to heterotrophic bacterial inactivation (Wang et al., 2017). In soil, heterotrophic bacteria are the main driver of soil carbon decomposition and greenhouse gas emission. As a result, the inactivation of heterotrophic bacteria results in protection of carbon from microbial degradation. Except for decomposition of soil organic carbon (SOC), the presence of HO• can also stabilize C in soil via a rapid formation of new intermolecular covalent bonds among soil components (Piccolo et al., 2011). Formation of new intermolecular covalent bonds increases the recalcitrance of SOC." (Pages 21-22, Lines 748-769 in the revised manuscript)

"The mobilized Fe can be easily transformed into the newly-formed reactive Fe (hydro)oxides (especially poorly crystalline Fe oxides) (Kleber et al., 2005; Yu et al., 2017), which will promote the formation of organo-mineral associations that are chemically more stable (Koegel-Knabner et al., 2008)." (Page 22, Lines 775-778 in the revised manuscript)

- [1.439-442]: "In addition, the generation of free radicals may also have indirect effects on bacterial growth via substrate availability (Table S4). Substrate availability is improved in the presence of radicals, owing to the following two facts: 1) the depolymerization role of radicals on the complex substrates; 2) the inhibition role of radicals on bacteria indirectly increasing the amounts of available substrates.": Do you think that we can see an inhibition of bacterial growth through the reduction of nutrient availability induced by free radicals in a medium where nutrients are in excess?

**R:** Our results did not confirmed an inhibition of bacterial growth through the reduction of nutrient availability induced by free radicals. In the revised manuscript, we deleted "the inhibition role of radicals on bacteria indirectly increasing the amounts of available substrates".

- What about the role of minerals on the "stabilization-adsorption" of organic compounds of the NB medium?

**R**: Minerals may interact with organic compounds of the NB medium, including adsorption (owing to a big specific surface area) and the formation of organo-mineral complexes (i.e., stablization). However, these interactions are not discussed/included in this manuscript.

- [1.445-448]: Fe is one of the numerous processes regulating carbon cycle in soils. I suggest something like: "In this study, we suggest that soil carbon cycle is partly regulated by Fe minerals (i) by the formation of organo-mineral complexes and (ii) by the bacterial development inhibition (specify the processes)."

**R:** Agree! In the revised manuscript, we changed "In this study, we suggest that soil carbon storage is regulated by Fe minerals, not only because of the formation of organo-mineral complexes (Kögel-Knabner, 2002; Kleber and Johnson, 2010; Schmidt et al., 2011) but also due to the bacterial inhibition activity of Fe minerals." (Page 20, Lines 445-448 in the original manuscript)

**to**

"In this study, we suggest that soil carbon cycle is partly regulated by Fe minerals (i) by the formation of organo-mineral complexes (Kögel-Knabner, 2002; Kleber and Johnson, 2010; Schmidt et al., 2011) and (ii) by the bacterial development inhibition." (**Page 22, Lines 778-781 in the revised manuscript**)

- 1.451 replace "but" by "and"

**R: Done.**

Conclusions

- [l. 467-458]: "effects on bacterial growth and the presence of minerals may potentially interfere with the measurement of cell numbers": I do not think that it is necessary to speak about that here.

**R:** In the revised manuscript, we deleted this sentence. The deleted part could be seen

in the tracked changes of Marked Manuscript and did not listed here for brevity.

**List of all relevant changes made in the manuscript**

**Title**

-1. Lines 1-3, "Bacteria" was changed to "Pseudomonas brassicacearumJ12".

**Abstract**

-2. Line 26, "*Pseudomonas* J12" was changed to "the model bacteria--Pseudomonas brassicacearum J12".

-3. Line 28, "Pseudomonas" was deleted.

-4. Line 30, "bacteria" was changed to "J12".

-5. Line 36 "bacteria" was changed to "the growth of *Pseudomonas brassicacearum* J12"

-6. Lines 37-39 "further contribute to soil carbon storage" was changed to "serve as an ubiquitous mechanism between iron minerals and all of the heterotrophic bacteria in view of taxonomically and ecologically diverse heterotrophic bacteria from terrestrial environments as a vast source of superoxide".

**Introduction**

-7. Lines 53-54 "such as increase of membrane permeability and oxidative damage" was added.

-8. Line 73 "heterotrophic" was added.

- -9. Line 74 "; Tang et al., 2013" was deleted.
- -10. Line 77 "oxidation" was changed to "reactions".
- -11. Lines 79-80 " $\rightarrow$ " was changed to " $\rightarrow$ ".
- -12. Lines 90-92 "heterotrophic" was added.
- -13. Line 98 "the model bacteria--" was added.

-14. Lines 103-104 "Specifically, montmorillonite and kaolinite are Al(III)-containing minerals, while hematite, goethite and ferrihydrite belong to Fe(III)-containing minerals." was added.

- -15. Line 105 "heterotrophic" was added.
- -16. Line 108 "bacterial" was deleted and "on J12" was added.

-17. Lines 109-120 "bacteria" was changed to "J12".

**Materials and Methods**

-18. Line 125 "Cultivation experiments" was changed to "Mineral preparation".

-19. Lines 126-128 "Al2O3·2SiO2·2H2O," and "(Al2,Mg3)Si4O10(OH)2·nH2O," were added.

-20. Line 140 "2.2. Pseudomanas cultivation experiments" was added.

-21. Line 141 "The bacterium used in this experiment is *Pseudomonas brassicacearum* J12, which is a major group of rhizobacteria that aggressively colonize plant roots, has been considered an important group for sustainable agriculture, and was provided by Dr. Zhou (Zhou et al., 2012)." was deleted.

-22. Lines 160-161 "Measurement of the OD600 on mineral suspension was shown in Table S1." was added.

-23. Line 161 "bacterial growth" was changed to "the growth of J12".

-24. Line 165 "Table S1" was changed to "Fig. S4".

-25. Line 165-169 "According to the data provided by manufacturers, the specific surface area of kaolinite and montmorillonite are ~40 and 800 m2 g-1, respectively. The synthesis of hematite, goethite and ferrihydrite were referred to the method from Schwertmann and Cornell (2007), and their specific surface area are approximately 30, 20, 200-300 m2 g-1, respectively." was added.

-26. Line 170 "2.2." was changed to "2.3.".

-27. Line177 "Fig.S4" was changed to "Fig.S5".

-28. Line 183 "2.3." was changed to "2.4.".

-29. Line 183 "2.4." was changed to "2.5.".

-30. Line 212 "2.5." was changed to "2.6.".

-31. Line 246 "2.6." was changed to "2.7.".

-32. Lines 256-259 "The pH of *Pseudomonas brassicacearum* J12 cultivated with different minerals or without mineral (control) was detected after 12 h. Eh of the suspension of minerals alone (25 mg ml-1) and of bacteria-mineral mixture was detected by redox potentiometer (Orion star A211, Thermo Fisher scientific, USA)." was added.

-33. Line 261 "2.7." was changed to "2.8.".

-34. Lines 265-269 "The one sample Kolmogorov-Smirnov test is used to test whether a sample comes from a specific distribution. In this study we used this procedure to determine whether the data set was normally distributed. In the regression equation, the parameters R and t represent coefficient of determination and the result of *t*-test." was added.

**Results**

**-35. Line 273 "**3.1. Bacterial inhibition by minerals**" was changed to "**3.1. Effect of mineral nature and their concentrations on J12 development**".**

-36. Line 274 "The effects of the nature and content of tested minerals on the  $OD_{600}$  of *Pseudomonas brassicacearum* J12 subcultures taken after 12 h growth are shown in Fig. 1." was deleted.

-37. Lines 275-277 "Compared to Control (i.e., no minerals), the presence of montmorillonite significantly increased  $OD_{600}$ ." was changed to "Specifically, the OD600 values of samples were  $0.43 \pm 0.01$ ,  $0.44 \pm 0.02$  and  $0.43 \pm 0.01$  at the concentration of 5, 10 and 25 mg mL-1, respectively."

-38. Lines 277-282 "On the other hand, presence of all other investigated minerals decreased OD600 in the following order: ferrihydrite > goethite > hematite > kaolinite at 5 and 25 mg mL-1, and ferrihydrite > goethite > kaolinite > hematite at 10 mg mL-1, suggesting that montmorillonite promoted the growth of *Pseudomonas brassicacearum* J12, but the rest of the tested minerals inhibited its growth." was changed to "Presence of all other investigated minerals decreased OD600 in the following order: ferrihydrite ( $0.24 \pm 0.04$  and  $0.09 \pm 0.01$ ) > goethite ( $0.26 \pm 0.02$  and  $0.14 \pm 0.00$ ) > hematite ( $0.30 \pm 0.03$  and  $0.16 \pm 0.02$ ) > kaolinite ( $0.16 \pm 0.02$ ) > goethite ( $0.18 \pm 0.02$ ) > kaolinite ( $0.21 \pm 0.02$ ) > hematite ( $0.28 \pm 0.02$ ) at 10 mg mL-1."

-39. Lines 282-284 "Meanwhile, an increase in mineral concentration resulted in a significant decrease in  $OD_{600}$ , except for montmorillonite, as the  $OD_{600}$  seemed to be independent of its concentration (Fig. 1)." was changed to "An increase in mineral concentration resulted in a significant (p < 0.05) decrease in  $OD_{600}$ . However, in
presence of montmorillonite the  $OD_{600}$  is stable at about 0.43 for all the mineral concentration studied."

-40. Line 285 "3.2. Chemical structure of minerals" was added.

-41. Line 355 "The Mn(II) in the montmorillonite was reported to act as a scavenger for any hydroxyl radical production(Garrido-Ramírez et al., 2010), which may explain the promotion of microbial growth." was deleted.

-42. Line 356 "3.2." was changed to "3.3.".

-43. Line 357 "Pseudomonas brassicacearum" was deleted.

-44. Lines 358-360 "almost similar to the control" was changed to "similar (p > 0.05) to the control at low concentration (i.e., 5 mg mL-1) but significant different (p < 0.05) at high concentration (i.e., 25 mg mL-1)".

-45. Line 364 "rapidly" was changed to "significantly".

-46. Line 366 "Fig.S5" was changed to "Fig. S6".

-47. Line 367 "3.3." was changed to "3.4.".

-48. Lines 368-482 the part of **Iron chemistry and its correlation with HO** and **OD**600 was reorganized.

-49. Line 483 "3.4." was changed to "3.5.".

-50. Lines 488-489 "Fig. 5a" was changed to "Fig. 7a".

-51. Lines 492-515 "(Spot B in Fig. 5b and Table S3). However, considerable percentages of hematite (~13%) and goethite (~19%) were present on the edge of these mineral particles (Spot A in Fig. 5b and Table S3)." was changed to "(Spot A in Fig. 7b and Table S3). However, considerable percentages of hematite (~13%), goethite (~19%) and FeC2O4 (~25.9%) were present on the edge of these mineral particles (Spot B in Fig. 7b and Table S3)."

-52. Line 516 "Fig. 5c" was changed to "Fig. 7c".

-52. Line 520 "good" was changed to "significant".

-53. Line 522 "Fig. S9" was changed to "Fig. S10".

-54. Lines 523-532 the part of 3.6. Effect of the presence of J12 on surface Fe

species (XPS analysis) was added.

**Discussion**

-55. Line 534 "bacterial" was changed to "J12".

-56. Lines 575 "bacterial growth" and "the bacterial growth" were changed to "the growth of J12" and "its", respectively.

-57. Line 582 "bacteria" was changed to "J12".

-58. Lines 586-588 "Furthermore, the association of a cell labeling with DAPI and a count of labeled cells with flow cytometry (or fluorescence microscopy) is also an alternative choose." was added.

-59. Line 589 "A previous study showed that a 58  $\mu$ M (~1.6 mg L-1) concentration of Al has toxicological effects on *Pseudomonas sp.* (Illmer and Schinner, 1999)." was deleted.

-60. Line 592 "bacteria (Fig. S8)" was changed to "J12 (Fig. 6)".

-61. Lines 592-597 "It is worth noting that >2 mg L-1 of aqueous Al(III) was detected for montmorillonite experiments with the passage of time (Fig. 6); however, the growth of J12 was not inhibited (Fig. 1). This may be attributed to the adsorption of aqueous Al(III) by bacterial EPS, which further protected bacteria from damage (Wu et al., 2014). However, direct evidence is lacked in this study and thus further investigation is needed to address this issue." was moved from Line 612 to Lines 592-597.

-62. Lines 617-619 "However, we did not detect a significant decrease of pH in this study (Fig. 4), suggesting that the formation of some Al intermediates may be slightly." was added.

-63. Line 620 "bacteria" was changed to "J12".

-64. Line 623 "bacterial inhibition efficiency" was changed to "inhibition efficiency on J12".

-65. Line 627 "heterotrophic" was added.

-66. Line 663 "bacteria" was changed to "J12".

-67. Lines 664-665 "bacterial inhibition activity" was changed to "inhibition activity of J12".

-68. Line 668 "Fig. S7" was changed to "Fig. S9".

-69. Lines 669-670 "which was" and "Therefore," were added.

-70. Line 681 "Fig. 4" was changed to "Fig. 5".

-71. Line 695 "Fig. 5" was changed to "Fig. 7".

-72. Lines 695-698 "High percentage of the less stable ferrihydrite (Table S3) may be attributable to the stabilization role of produced EPS (Fig. 5c) by bacteria to minerals, which had been shown during the cultivation of fungi with minerals (Li et al., 2016)." was changed to "High percentage of the less stable ferrihydrite (Table S3) may be attributable to the stabilization role of produced EPS (Fig. 7c) by J12 to minerals. It is consistent with a previous finding in the cultivation of fungi with minerals (Li et al., 2016)."

-73. Lines 698-701 "The stabilization role of EPS on meta-stable ferrihydrite was mainly identified as its combination into the network structure of minerals, which prevents the formation of crystalline minerals (Braunschweig et al., 2013)." was added.

-74. Line 708 "Fig. 6" was changed to "Fig. 8".

-75. Lines 710-712 "Fe(II) can also be generated by catalyzing a series of cellular intracellular (e.g., glutathione and NAD(P)H) and free (e.g., L-cysteine and FADH2) reductants (Imlay, 2003)." was changed to "Fe(II) can also be generated by catalyzing a series of intracellular reductants (e.g., glutathione, NAD(P)H, L-cysteine and FADH2) (Imlay, 2003).".

-76. Line 734 "bacterial inhibition activity" was changed to "the inhibition activity of J12".

-77. Line 708 "Fig. S7" was changed to "Fig. S9".

-78. Lines 735-737 "Intracellular oxidative toxicity caused by soluble Fe(III) played an important role in bacterial inhibition activity simultaneously (Schoonen et al., 2006)." was changed to "Intracellular oxidative toxicity also caused by soluble Fe(III) played an important role in the inhibition activity (Schoonen et al., 2006)."

-79. Line 737 "bacteria" was changed to "J12".

-80. Lines 740-742 "bacterial" was changed to "J12".

-81. Line 743 "Fig. 7" was changed to "Fig. 9".

**-82. Line 745 "extracellular" was added.**

-83. Lines 746-772 "Oxidative damage of extracellular HO' may lead to bacterial inactivation, and protection of carbon from microbial degradation. In addition, the generation of free radicals may also have indirect effects on bacterial growth via substrate availability (Table S4). Substrate availability is improved in the presence of radicals, owing to the following two facts: 1) the depolymerization role of radicals on the complex substrates; 2) the inhibition role of radicals on bacteria indirectly increasing the amounts of available substrates." was changed to "Soluble Fe(II) and Fe(III) released from minerals can penetrate into the cell membranes, thereby inducing intracellular oxidative damage (Williams et al., 2011). Oxidative damage of HO' may induce the damage of a membrane lipid and cardiolipin that can lead to heterotrophic bacterial inactivation (Wang et al., 2017). In soil, heterotrophic bacteria are the main driver of soil carbon decomposition and greenhouse gas emission. As a result, the inactivation of heterotrophic bacteria results in protection of carbon from microbial degradation. Except for decomposition of soil organic carbon (SOC), the presence of HO' can also stabilize C in soil via a rapid formation of new intermolecular covalent bonds among soil components (Piccolo et al., 2011). Formation of new intermolecular covalent bonds increases the recalcitrance of SOC. In addition, the generation of free radicals may also have indirect effects on J12 growth via substrate availability (Table S4). Substrate availability is improved in the presence of radicals, owing to the depolymerization role of radicals on the complex substrates.".

-84. Line 773 "soil organic carbon" was deleted.

-85. Lines 775-778 "The mobilized Fe can be easily transformed into the newly-formed reactive Fe (hydro)oxides (especially poorly crystalline Fe oxides) (Kleber et al., 2005; Yu et al., 2017), which will promote the formation of organo-mineral associations that are chemically more stable (Koegel-Knabner et al., 2008)." was added.

-86. Lines 778-781 "In this study, we suggest that soil carbon storage is regulated by Fe minerals, not only because of the formation of organo-mineral complexes (Kögel-Knabner, 2002; Kleber and Johnson, 2010; Schmidt et al., 2011) but also due to the bacterial inhibition activity of Fe minerals." was changed to "In this study, we suggest that soil carbon cycle is partly regulated by Fe minerals (i) by the formation of organo-mineral complexes (Kögel-Knabner, 2002; Kleber and Johnson, 2010; Schmidt et al., 2011) and (ii) by the bacterial development inhibition.".

-87. Line 783 "*Pseudomonas brassicacearum*" was deleted and "but" was changed to "and".

**Conclusions**

-88. Line 789 "brassicacearum" was added.

-89. Lines 812-814 "Pseudomonas" was deleted.

-90. Line 818 "The generation of HO' by Fe(III)-containing minerals follows the order ferrihydrite > goethite > hematite. In addition, the generation of free radicals may also have indirect (i.e., substrate availability) effects on bacterial growth and the presence of minerals may potentially interfere with the measurement of cell numbers." was deleted.

-91. Line 819 "heterotrophic" was added.

-103. Line 1059 "Figure 5" was changed to "Figure 7".

-104. Lines 1063-1066 "Red color in (a) represents high density of Fe, followed by orange, yellow, green, little green, and purple. Red color in (c) indicates the highest intensity of functional groups, followed by yellow, green, and blue." was added.

**-105. Line 1068 "Figure 6" was changed to "Figure 8".**

-106. Lines 1076-1078 "In subfigure (b) and (c), dark, orange and other lines represents the raw spectrum, fitted spectrum and the component of fitted Fe species." was added.

-107. Lines 1079-1084 "Figure 7. Schematic of the bacterial inhibition by Fe(III)-containing minerals through a free-radical mechanism." was changed to "Figure 9. Schematic of the inhibition of heterotrophic bacteria by Fe(III)-containing minerals through a free-radical mechanism.  $\mathbb{O}$ -④ represent the processes occurring at heterotrophic bacterial-mineral interfaces and are detailed in the main text.  $\mathbb{O}$  Production of HO' through the Fenton or Fenton-like reactions; ② Direct inhibition of heterotrophic bacteria by HO'; ③ Indirect inhibition of heterotrophic bacteria by HO'; ④ Intracellular inhibition of heterotrophic bacteria by soluble Fe.".

-108. Line 1090 The revised Fig.1.

-109. Lines 1093-1096 "K, kaolinite; M, montmorillonite; H, hematite; G, goethite;

[revised manuscript text omitted]